# Cryo-ET and MD simulations reveal that dynein-2 is tuned for binding to the A-tubule of the ciliary doublet

Haoqiang K He [1,4], Shintaroh Kubo [2,4], Xuwei Chen [1,4], Qianru H Lv [1], Azusa Kage [3] & Muneyoshi Ichikawa [1✉]

## Abstract

Eukaryotic cilia and flagella are thin structures present on the surface of cells, playing vital roles in signaling and cellular motion. Cilia assembly depends on intraflagellar transport (IFT) along doublet microtubules (doublets). Unlike dynein-1, which works on cytoplasmic singlet microtubules, dynein-2 works on the doublets inside cilia. Previous studies have shown that retrograde IFT, driven by dynein-2, occurs on the A-tubule of the doublet, suggesting an elusive mechanism by which dynein-2 recruits retrograde IFT to the A-tubule. Here, we investigated the molecular basis of this mechanism using cryo-electron tomography (cryo-ET), molecular dynamics (MD) simulations, and biochemical analysis. Our biochemical assays revealed that the microtubule-binding domain of dynein-2 exhibits a higher affinity for the ciliary doublets than dynein-1. Cryo-ET further visualized the preferential binding of dynein-2 to the A-tubule of the doublet. MD simulations suggest that dynein-2 prefers the tyrosinated tubulin lattice as is present in the A-tubule. These findings reveal a recruitment mechanism of retrograde IFT by dynein-2, providing new insights into the spatial and functional specialization of ciliary doublets.

**Keywords** Cilia; Cryo-ET; Dynein-2; IFT; MD Simulations
**Subject Categories** Cell Adhesion, Polarity & Cytoskeleton; Membranes & Trafficking; Structural Biology

## Introduction

In eukaryotic cells, cilia (also referred to as flagella) are highly conserved hair-like organelles that extend from the cell surface and have a wide range of functions. Non-motile primary cilia play roles in signaling, and motile cilia provide motility for specific cells like sperm. Both types of cilia possess nine doublet microtubules (doublets) composed of a completely cylindrical A-tubule with 13

protofilaments (PFs) and an incompletely cylindrical B-tubule with 10 PFs (Gibbons, 1981; Ishikawa and Marshall, 2011) (Fig. 1). The B-tubule is assembled on the outside of the A-tubule, and some PFs of the A-tubule are covered by the B-tubule. The ciliary doublets are decorated by, or interact with various microtubule-associated proteins (MAPs) for stabilization, as well as protein complexes involved in ciliary beating, such as axonemal dyneins, radial spokes, and the nexin-dynein regulatory complex. The PFs of the doublet facing the membrane around PFs-A8, A9, and B1-B4 are not covered, and this space is used as a railway for intraflagellar transport (IFT) (Leung et al, 2025; Walton et al, 2023). The IFT is a cargo transport process that takes place in the space between doublets and ciliary membranes and plays a vital role in the assembly and maintenance of the complex structure of cilia (Webb et al, 2020). The IFT complex consists of two sub-complexes, IFT-A and IFT-B, and IFT proteins are linked to various human ciliopathies (Reiter and Leroux, 2017). These sub-complexes assemble sequentially to form IFT trains, which are transported along doublets by the motor proteins dynein-2 and kinesin-2 (Fig. 1).

Dynein-2 is a multiprotein complex comprising two heavy chains (HCs), a heterodimer of intermediate chains (ICs) (WDR60/WDR34), two light intermediate chain-3 (LIC3), homodimeric light chain (LC) DYNLRB, three homodimeric LC8, and heterodimeric LC (TCTEX/TCTEX1D2) (Mukhopadhyay et al, 2024; Toropova et al, 2019) (Fig. 2A). Dynein-2 and its cytoplasmic counterpart, dynein-1, share a similar domain organization, including a tail domain and a head domain, which is composed of the AAA+ motor domain, coiled-coil stalk, and microtubule-binding domain (MTBD). Dyneins interact with microtubule (MT) with MTBD that undergoes the transition of MT low-affinity and MT high-affinity states (Gibbons et al, 2005; Redwine et al, 2012). Despite the similarity, their functions are distinct: dynein-1 operates on singlet MTs (singlets) in the cytoplasm, whereas dynein-2 is specialized for transportation on ciliary doublets (Yildiz and Zhao, 2023).

In the IFT process, the anterograde IFT train driven by kinesin-2 transports cargo and dynein-2 from the cytoplasm to the tip of the cilia. Then, the anterograde IFT train remodels into the retrograde IFT train and transports back toward the cell body by

[1]State Key Laboratory of Genetics and Development of Complex Phenotypes, Department of Biochemistry and Biophysics, School of Life Sciences, Fudan University, Shanghai, China. [2]Department of Applied Chemistry, Graduate School of Engineering, The University of Tokyo, Tokyo, Japan. [3]Graduate School of Engineering, Muroran Institute of Technology, Muroran, Hokkaido, Japan. [4]These authors contributed equally: Haoqiang K He, Shintaroh Kubo, Xuwei Chen. ✉E-mail: ichikawa_muneyoshi@fudan.edu.cn

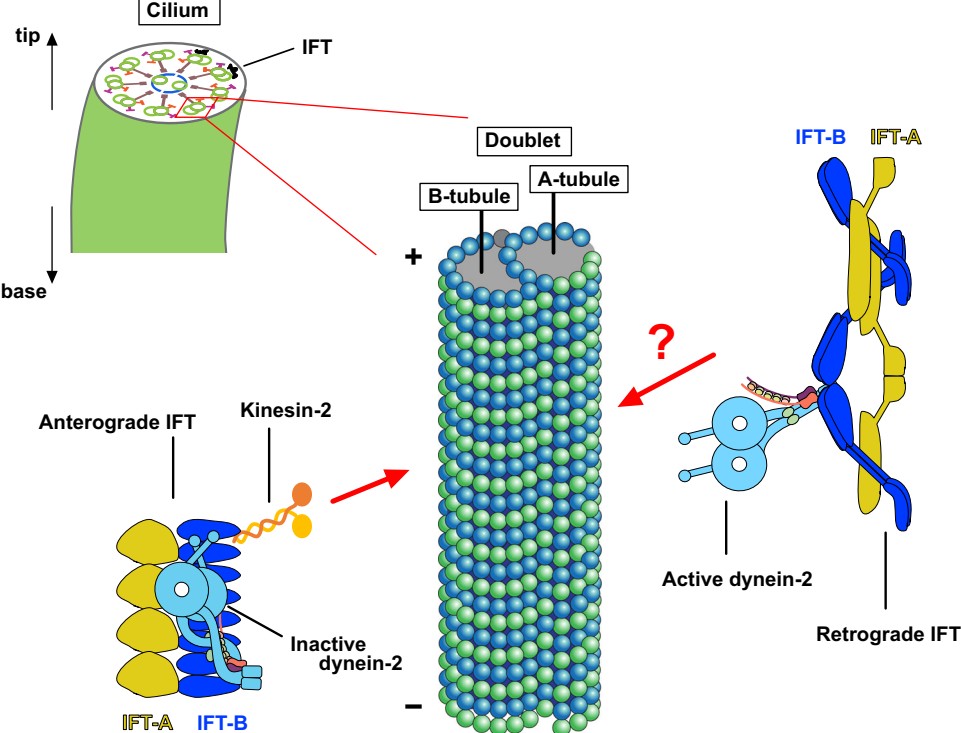

**Figure 1. Schematic illustration of the structure of the cilia and the mechanism of the IFT.**

Dynein-2 drives retrograde IFT toward the ciliary base on A-tubules, while kinesin-2 powers anterograde IFT toward the ciliary tip on B-tubules. In the anterograde IFT, dynein-2 takes a stacked conformation, and MTBDs are not associated with doublets. Microtubule polarity is indicated by (+) and (−).

dynein-2 (Lacey et al, 2024) (Fig. 1). While being transported by kinesin-2, dynein-2 adopts an autoinhibited conformation and is stored in the IFT train (Jordan et al, 2018; Toropova et al, 2019). To avoid the collision of anterograde and retrograde IFT trains in the limited space between the doublet and the membrane, the doublet serves as a dual railway track for IFT. From the previous electron tomography study of resin-embedded flagella, anterograde IFT trains were found on the B-tubule, while retrograde IFT trains were found on the A-tubule (Stepanek and Pigino, 2016).

Since the tubulins composing A- and B-tubules have different post-translational modifications (PTMs) like tyrosination/detyrosination, polyglutamylation, and polyglycylation (Westermann and Weber, 2003), the different PTMs could lead to the sorting of the anterograde and retrograde IFTs. A recent study has shown that polyglycylation is present in both A- and B-tubules (Alvarez Viar et al, 2024) and, therefore, cannot be the cause for sorting the IFTs. The same study located the polyglutamylation on the PF-B9 of the B-tubule that is not the track of IFT.

The tyrosination/detyrosination occurs on the C-terminus of the α-tubulin (Sanyal et al, 2023). In most cases, expressed α-tubulin has a tyrosine residue at the C-terminus (tyrosinated), and this tyrosine residue can be enzymatically removed by tubulin carboxypeptidases (TCPs) (detyrosination). The removal of the C-terminal tyrosine residue exposes the glutamic acid residue at the C-terminus. A tyrosine residue can be re-attached to the detyrosinated tubulin by the tubulin tyrosine ligase (TTL). The A-tubule is shown to be enriched with tyrosinated tubulins, and the B-tubule has detyrosinated tubulins (Johnson, 1998). Very recently,

it was found that in the *VashL Chlamydomonas* mutant, the lack of tubulin detyrosinase led to frequent collisions of anterograde and retrograde IFT trains, meaning that the tyrosination/detyrosination balance of the doublet was important for the proper sorting of IFT trains (Chhatre et al, 2025). Despite these advances, the exact mechanism by which retrograde IFT is selectively recruited to the A-tubule remains unidentified.

Here, we used biochemical analysis and showed that dynein-2 exhibits higher affinity for doublets than dynein-1. By cryo-electron tomography (cryo-ET) analysis aided by Volta Phase Plate (VPP) of an in vitro reconstitution system, we found that dynein-2 prefers to bind to the A-tubule of the doublet. Our molecular dynamics (MD) simulations suggest that dynein-2 favors a tyrosinated tubulin lattice.

## Results

### Dynein-2 is tuned for binding to the ciliary doublet

We hypothesized that the microtubule-binding properties of dynein-2 differ from those of dynein-1, reflecting the distinct intracellular functional locations. To verify this, we compared available MTBD structures (PDB IDs: 5AYH and 4RH7) (Nishikawa et al, 2016; Schmidt et al, 2015) and sequences of dynein-2 and dynein-1 (Fig. 2B,C). The overall structures of the MTBDs were highly similar, with an RMSD of 1.1 Å for the MTBD region. Some residues of the *Dictyostelium* dynein that were previously

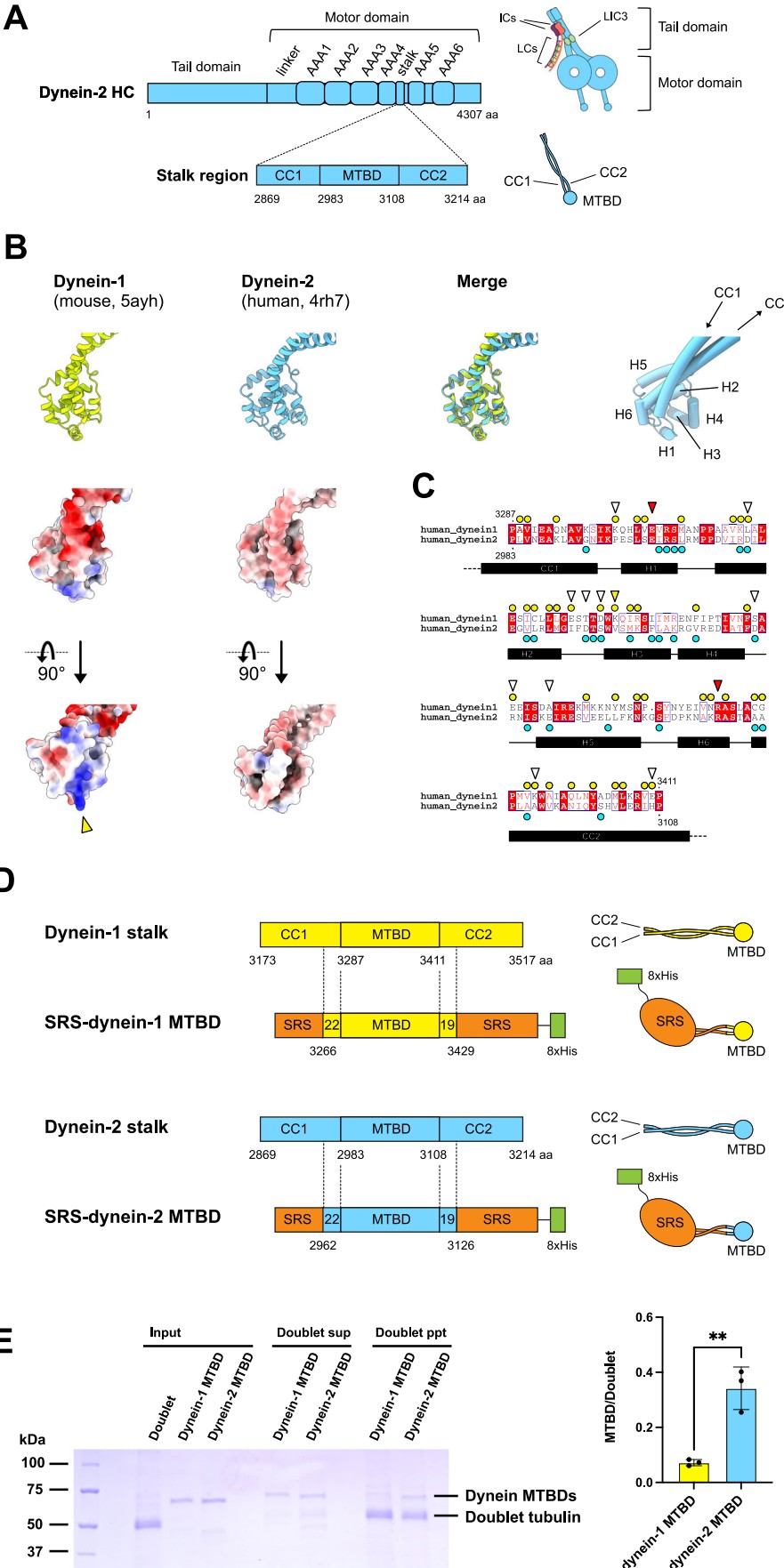

◀ **Figure 2. Comparison of MTBDs of dynein-1 and dynein-2.**

(A) Schematic representation of the dynein-2 HC and stalk region and their structures. The HC includes the tail domain and the motor domain. The stalk region of the motor domain is highlighted, and CC1, MTBD, and CC2 within the stalk region are indicated. The diagram includes subunits associated with the dynein-2 tail domain. (B) Comparison of the dynein-1 MTBD and dynein-2 MTBD structures. Dynein-1 stalk structure (top row, leftmost, PDB ID: 5AYH) (Nishikawa et al, 2016) and dynein-2 motor domain structure (top row, second, PDB ID: 4RH7) (Schmidt et al, 2015) are aligned based on the MTBD region. Matched structures (top row, third) show that the general structures of MTBD of dynein-1 and dynein-2 are similar. Surface models of dynein-1 and dynein-2 MTBDs (second row), colored to show electrostatic potential ranging from red (−10 kcal/(mol·e)) to blue (+10 kcal/(mol·e)). 90° rotated views of surface models (bottom row) show that the contact surfaces of dynein-1 and dynein-2 MTBDs to the tubulin lattice are distinct. Helix numbers, CC1, and CC2 are indicated in the rightmost panel. The yellow arrowhead indicates the K3336 residue of H3 from dynein-1 (the residue number is based on human dynein-1). (C) Sequence alignment of human dynein-1 and human dynein-2 MTBD regions. Amino acid sequences of dynein-1 MTBD region (NCBI Reference Sequence: NP_001367.2) and dynein-2 MTBD region (NCBI Reference Sequence: NP_001368.2) are aligned. The alignment was performed using Clustal W (Thompson et al, 1994), and the figure was generated using ESPript 3.0 (Robert and Gouet, 2014). Note that the sequences of the mouse dynein-1 MTBD region (PDB ID: 5AYH) and the human dynein-1 MTBD region are exactly the same. The yellow circles indicate the amino acid residues conserved between dynein-1, and the cyan circles show the amino acid residues conserved between dynein-2, respectively (see also Fig. EV1). K3336 of dynein-1 is shown by the yellow arrowhead. Red arrowheads indicate the residues previously shown as important in MT binding. Positions of CC1, CC2, and helices are indicated at the bottom. The residues conserved either in dynein-2 or dynein-1 are indicated by white arrowheads (see also Fig. EV1). (D) Schematic diagrams of the stalk regions and the MTBD constructs used in this study. CC1 and CC2 of the MTBD regions were merged with SRS coiled-coil (orange) to fix MTBD structures to MT strong binding conformations, and 8×His-tag (green) was added for purification. (E) Co-pelleting assay of dynein MTBD constructs with doublets. MTBD constructs were incubated with doublets, ultracentrifuged, and analyzed by SDS-PAGE (left), and the bands were quantified (right). The band of the dynein-2 MTBD construct in the precipitation (ppt) fraction was more prominent compared with that of dynein-1 MTBD construct. Mean values from three independent experiments are shown in the graph, and error bars represent the SD. Statistical analyses were performed using an unpaired $t$ test (** indicates $P \leqq 0.01$, $P = 0.0039$). Source data are available online for this figure.

shown to form salt bridges with the tubulin residues (Uchimura et al, 2015) were conserved in human dynein-2 and dynein-1 HCs. For instance, E3002 of H1 from dynein-2 (equivalent to E3306 of dynein-1 and E3390 from *Dictyostelium* dynein) and R3081 of H6 from dynein-2 (equivalent to R3384 of dynein-1 and R3469 from *Dictyostelium* dynein) (Fig. 2C, red arrowheads). Despite these similarities, the surface charge of the MTBD in dynein-2 was notably distinct from that in dynein-1 (Fig. 2B). In dynein-1, the MTBD displays a prominent patch of positive charge at the MT-binding interface, whereas dynein-2 shows a more neutral, and even slightly negatively charged surface in the corresponding region. A key difference was the absence of K3336 from H3 of dynein-1 in the dynein-2 MTBD (Fig. 2B,C, yellow arrowheads). Other residues conserved within dynein-2 MTBD were also distinct from those in dynein-1 MTBD (white arrowheads in Figs. 2C and EV1). For instance, a conserved lysine residue of dynein-1 between CC1 and H1 (K3301 in human) is replaced by proline residue in dynein-2 (P2997 in human). A conserved leucine residue of dynein-1 (L3319 in human) located in H2 is also changed to aspartic acid in dynein-2 in several species (D3015 in human). In addition, a conserved aspartic acid (D3334 in human dynein-1) just before H3 is replaced by a serine residue in dynein-2 in multiple species. Collectively, these substitutions result in the more neutral surface charge of the dynein-2 MTBD.

To further explore these findings, we performed an MT co-pelleting assay using MTBD constructs and native doublets. We used the MTBD construct fused with seryl-tRNA synthase (SRS) for dynein-2 and dynein-1, respectively (Fig. 2D). The constructs used in this study are equivalent to the α registry (MT-high affinity) 22:19 construct in (Carter et al, 2008). For doublets, we purified the native doublets from *Tetrahymena* cilia and removed the associated proteins (Fig. EV2). Our previous structural analyses have shown that doublets containing Microtubule Inner Proteins (MIPs) but without MAPs outside can be prepared with this procedure (Ichikawa et al, 2019; Ichikawa et al, 2017). After incubating dynein MTBD constructs with doublets, the bound dynein MTBD proteins were pelleted with doublets by centrifugation and analyzed by SDS-PAGE. As a result, dynein-2 exhibited higher binding

compared with dynein-1 (Fig. 2E), consistent with dynein-2's function on doublets.

## Dynein-2 preferentially binds A-tubules

To further explore how dynein-2 interacts with the doublet, we exploited an in vitro reconstitution system constituted of minimal components. GST-Dyn2, lacking a tail domain and having mutations to mimic active dynein-2 with two heads open (Toropova et al, 2017) (Fig. 3A), was incubated with native doublets with clean outer surfaces as purified above (Fig. 3B). After the centrifugation, GST-Dyn2 was detected in the pellet fraction together with the doublets, and as the concentration of GST-Dyn2 increased, the binding was saturated (Fig. 3C).

The sample was vitrified and observed under saturated conditions by cryo-electron microscopy (cryo-EM). Because of the preferred orientation, the doublets were primarily observed in side views, with both tubules visible. From the obtained single tilt cryo-EM image, GST-Dyn2 molecules were accumulated on one side of the doublet (Fig. 3D). To further identify whether GST-Dyn2 binds more on the A- or B-tubule sides, we performed cryo-ET analysis aided by the Volta Phase Plate (VPP) (Fukuda et al, 2017; Imhof et al, 2019; Pöge et al, 2021) to enhance contrast and visualize the dynein-2 motor domains clearly (Fig. 3B; Movies EV1 and EV2). In the reconstructed tomograms, the A- and B-tubules of the doublets were readily distinguished both by the morphology of the tubules and the MIPs distinctive for the A- or B-tubule (Figs. 3E and EV4A).

By identifying the A- and B-tubules of the doublet, GST-Dyn2 molecules were found predominantly on the A-tubule side without external components (Fig. 3F). Comparison with available dynein-2 head structure (PDB ID: 4RH7) (Schmidt et al, 2015) confirmed that the GST-Dyn2 molecules are canonically bound to the doublets with its MTBDs (Fig. 3G). By quantifying the head numbers of GST-Dyn2 per 100 nm, the A-tubules had significantly more dynein-2 heads compared with the B-tubules (Fig. 3H). Given that PFs like A1, A10-13 in the A-tubule are inaccessible to dynein-2, while the PFs in the B-tubule are fully accessible, the actual

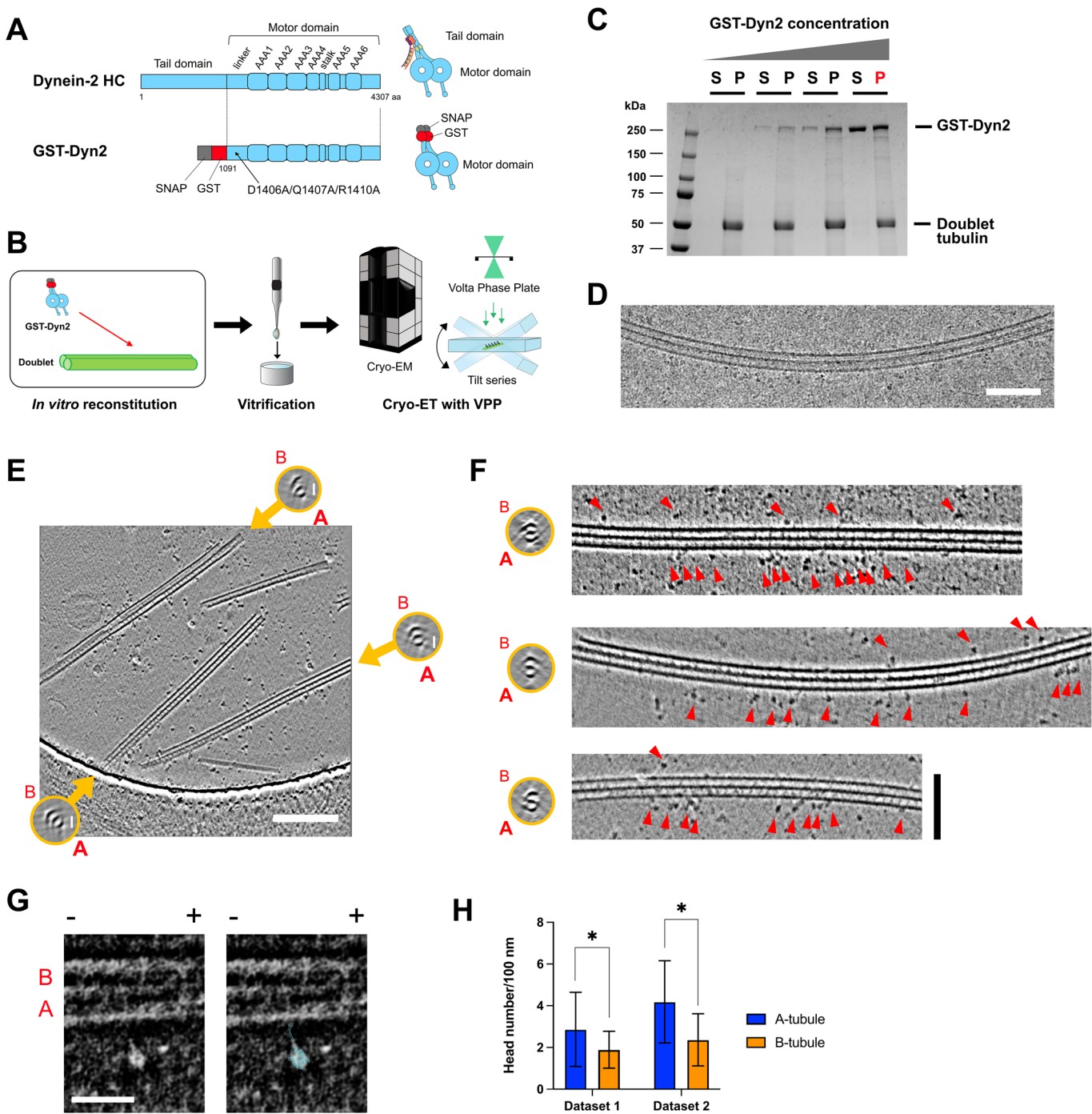

difference in binding is likely greater than what is observed. The preferential binding of GST-Dyn2 to the A-tubule was confirmed in another dataset, as well as using doublets from *Chlamydomonas* flagella (Figs. 3H, dataset 2, EV3, and EV4).

To obtain insights into the binding scheme of dynein-2 to the tubulin lattice, we analyzed the configuration of GST-Dyn2 molecules bound to the doublets. Three distinct configurations of the dynein-2 heads were observed, classified as Configuration 1, Configuration 2, and Configuration 3 (Fig. 4A). These configurations are defined by the spatial relationship between the two heads:

in Configuration 1, the head on the right-hand side (relative to the direction of movement) is leading; in Configuration 2, the two heads are aligned side by side; and in Configuration 3, the right-side head is trailing. Quantification showed that Configuration 3 was the majority (30.2%) compared with Configuration 1 (16.3%) or Configuration 2 (25.6%) (Fig. 4B). To test if observed configurations of GST-Dyn2 molecules reflect the physiological arrangement, we then performed the cryo-ET analysis of the WT *Chlamydomonas* flagella. The dynein-2 molecule with a similar configuration was observed within the flagella, supporting the

**Figure 3. Cryo-ET analysis of GST-Dyn2 bound to doublets.**

(A) Schematic illustrations of the dynein-2 HC and GST-Dyn2 construct. The tail domain of the dynein-2 HC was truncated, and the GST-tag (red) was added for dimerization with the SNAP-tag (gray). Mutations (D1406A/Q1407A/R1410A) were introduced to separate two motor domains. (B) The analysis workflow of the binding mode of the GST-Dyn2 to doublets. GST-Dyn2 was incubated with doublets and vitrified. The grid was loaded into cryo-EM, and tilt series were acquired using the VPP. (C) The SDS-PAGE image of the co-pelleting assay of GST-Dyn2 and doublets. A fixed concentration of doublets (500 μg/ml) was incubated with increasing concentrations of GST-Dyn2 (0, 112.5, 225, and 450 μg/ml), centrifuged, and the supernatant fraction (S) and precipitation fraction (P) were analyzed by SDS-PAGE. The precipitated sample (450 μg/ml, indicated in red) was resuspended in a buffer and used for vitrification. (D) A typical cryo-EM image of GST-Dyn2 decorated doublet microtubule. The image was taken with a single tilt with a nominal −2 μm defocus. There are more GST-Dyn2 bound to one side of the doublet. Scale bar, 100 nm. (E) Example of a tomographic slice of reconstructed GST-Dyn2 decorated doublets. Cross-sectional views are shown for each doublet in orange circles, and A- and B-tubules are indicated. Note that all three doublets in the tomogram have more GST-Dyn2 molecules on the A-tubule sides. There are several singlets due to sample preparations. Scale bar, 200 nm. See also Movies EV1 and EV2. (F) Representative examples of tomograms of GST-Dyn2 bound predominantly on the A-tubules of the doublets. Cross-sectional views (left, orange circles) and longitudinal views (right) are shown for each doublet, and A- and B-tubules are indicated. All doublet structures are shown in the figure so that the A-tubule sides face the bottom. Red arrowheads indicate the dynein-2 motor domains. Scale bar, 100 nm. (G) A tomographic slice showing a characteristic dynein motor domain bound to the doublet. Proteins are shown in white. The available dynein-2 motor domain structure (PDB ID: 4RH7) (Schmidt et al, 2015) is shown as interpretation in the right panel. A- and B-tubules are indicated. Polarity of the microtubule is shown by + and −. Scale bar, 50 nm. (H) Quantification of dynein-2 heads per 100 nm distance. Mean values of dynein-2 head numbers on either A- or B-tubules from two different cryo-ET datasets are plotted. Error bars represent the SD. Statistical analyses were performed using an unpaired t test (* indicates P≦0.05). Dataset 1 is same with the ones shown in (E–G). Typical slices of tomograms from dataset 2 are shown in Fig. EV4B. For dataset 1, total 882 dynein-2 heads from 24 doublets were quantified, and P value was 0.0201. For dataset 2, total 492 dynein-2 heads from 11 doublets were quantified, and P value was 0.0130. Source data are available online for this figure.

physiological relevance of the observed binding modes of GST-Dyn2 (Fig. 4C).

## A subset of α-tubulins is tyrosinated in *Chlamydomonas* doublets

Previously, it has been shown by immunogold labeling that the A-tubule of the *Chlamydomonas* doublet is enriched with tyrosinated tubulins while the tubulins forming the B-tubule are detyrosinated (Johnson, 1998). To estimate the overall levels of tubulin tyrosination and detyrosination, we purified microtubule fractions from *Chlamydomonas* flagella and performed mass spectrometry (MS) analysis (Fig. EV3A–C). In the WT *Chlamydomonas* doublet, both tyrosinated α-tubulin C-terminal peptides (DFEEVGAESAEGAGEGEGEEY and EDLAALEKDFEEVGAE-SAEGAGEGEGEEY) and the detyrosinated peptide (EDLAA-LEKDFEEVGAESAEGAGEGEGEE) were detected. In contrast, in *VashL* mutant *Chlamydomonas*, which lacks tubulin detyrosinase, only tyrosinated α-tubulin peptides (DFEEVGAESAEGAGEGE-GEEY and EDLAALEKDFEEVGAESAEGAGEGEGEEY) were detected (Fig. EV3D). In the WT sample, tyrosinated α-tubulin constituted 33.7% of the total detected C-terminal peptides, while detyrosinated α-tubulin comprised 66.3%, despite the A-tubule having more PFs (13 PFs) than the B-tubule (10 PFs).

To further confirm our observations, we also analyzed the level of glutamylation (Fig. EV3E), and only 6.35% of the α-tubulin peptides were glutamylated. This is consistent with the previous report showing that only one PF undergoes glutamylation (Alvarez Viar et al, 2024). Taken together, these results suggest that a subset of PFs in the A-tubule, possibly the pathway of the retrograde IFT, are tyrosinated (Fig. EV3F, see also "Discussion").

## Dynein-2 prefers the tyrosinated tubulin lattice

To test if the dynein-2's binding is affected by the tyrosination/detyrosination states of the doublet tubulin lattice, we performed MD simulations. Two dynein-2 heads, one with an MT low-affinity state and the other with an MT high-affinity state, representing a dynein-2 dimer, were placed on the tubulin lattice (Figs. 5A and EV5A). The three initial configurations used in the simulations, namely Position 1, Position 2, and Position 3, were defined based on the head configurations observed in our cryo-ET analysis of GST-Dyn2 bound to the doublets (Fig. 4A,B). To observe the initial diffusional motion of the low-affinity state MTBD as it shifts from the high-affinity state to the low-affinity state, the low-affinity state MTBD was placed at the MT-binding site (see "Methods" for details). The trajectories of the low-affinity state dynein-2 head were simulated on the tubulin lattices with or without tyrosination by coarse-grained MD (Fig. 5B). After the calculation, the probabilities of the existence of low-affinity MTBD were plotted for tyrosinated and detyrosinated tubulin lattice, respectively (Figs. 5C,D, and EV5B,C). The heatmaps were generated for each initial position, and the sum of the three initial positions was plotted. In addition to the probability heatmaps, we also calculated the overflow proportion, defined as the percentage of trajectories in which the MTBD either detaches from the tubulin lattice or moves outside the simulated $3 \times 3$ lattice region.

In the sum result, dynein-2 MTBD tended to stay in the same position (1, 1) in the tyrosinated tubulin lattice compared with the detyrosinated tubulin lattice (Fig. 5C,D). Furthermore, overflow proportion, which includes the detachment from the tubulin lattice, was higher in the detyrosinated tubulin lattice (20.09%) compared with the tyrosinated tubulin lattice (10.70%), suggesting that dynein-2 molecules are easier to detach from the detyrosinated tubulin lattice.

Notably, starting from initial position 3 under detyrosination, the low-affinity leading head tended to diffuse ahead of the high-affinity trailing head (Fig. EV5D). When the trailing head transitions to the low-affinity state, the dynein-2 dimer will not move toward the minus end since the next binding site on the same PF is unavailable. This behavior was uncommon on tyrosinated tubulin lattices.

To gain further insight into the molecular basis of dynein-2's distinct binding behavior on tyrosinated or detyrosinated tubulin lattices, we examined residue-level contacts between MTBD and tubulin in the presence or absence of the C-terminal tyrosine in our MD simulations (Fig. 5E,F). In the detyrosinated condition, the

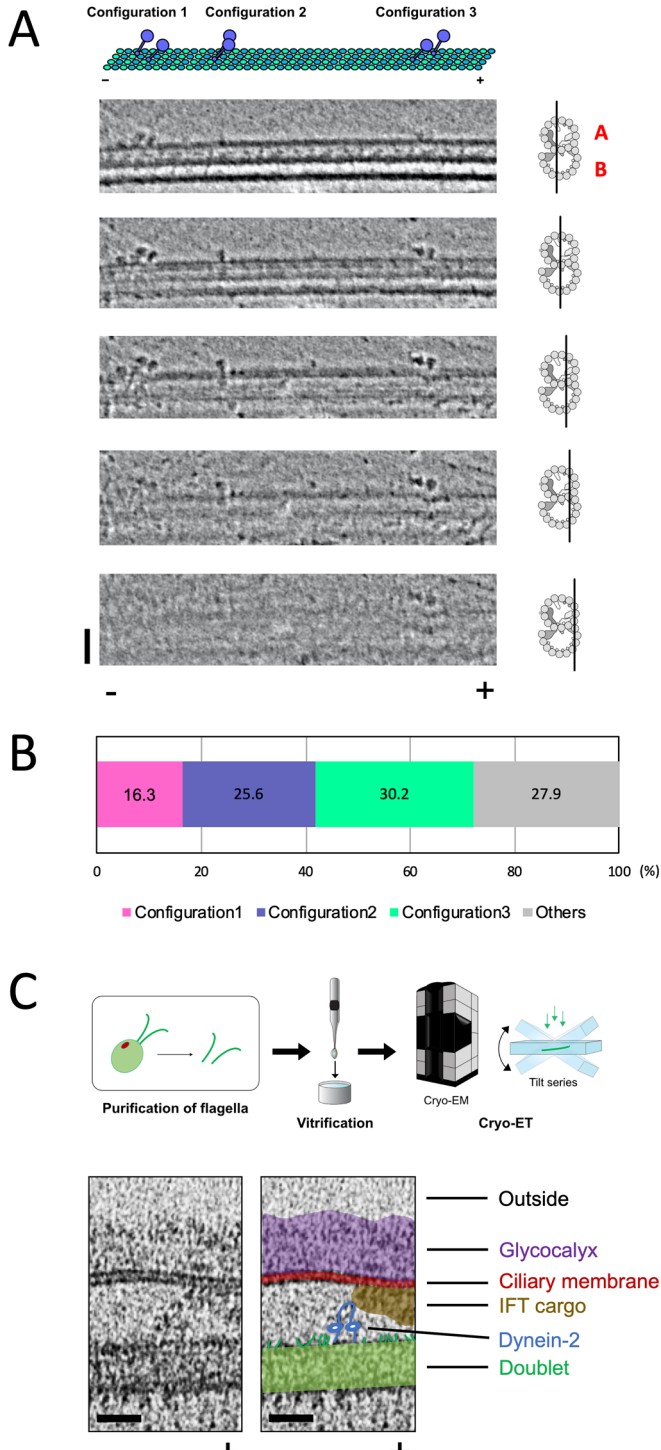

**Figure 4. Cryo-ET analysis of dynein-2 dimer configurations.**

(**A**) Sequential tomographic slices showing three distinct configurations of GST-Dyn2 dimers bound to doublets. Longitudinal slices with different depths are shown left. Black lines in the schematics of the transverse section (right) indicate the depths. A schematic diagram of interpretation is shown top. Scale bar, 50 nm. (**B**) Quantification of the three configurations of GST-Dyn2 on doublets. A total of 43 GST-Dyn2 molecules from five doublets were analyzed. Molecules with one head detached or in an unclear configuration are included in Others. (**C**) Cryo-ET result of *Chlamydomonas* flagella showing active dynein-2 in vivo. Flagella were purified, vitrified, and tilt-series were acquired (top). A representative tomographic slice from the reconstructed tomogram (bottom, left) and color-coded interpretation (bottom, right): glycocalyx (purple), ciliary membrane (red), IFT cargo (brown), dynein-2 (blue), doublet (green). + and - indicate microtubule polarity. Scale bar: 25 nm. Source data are available online for this figure.

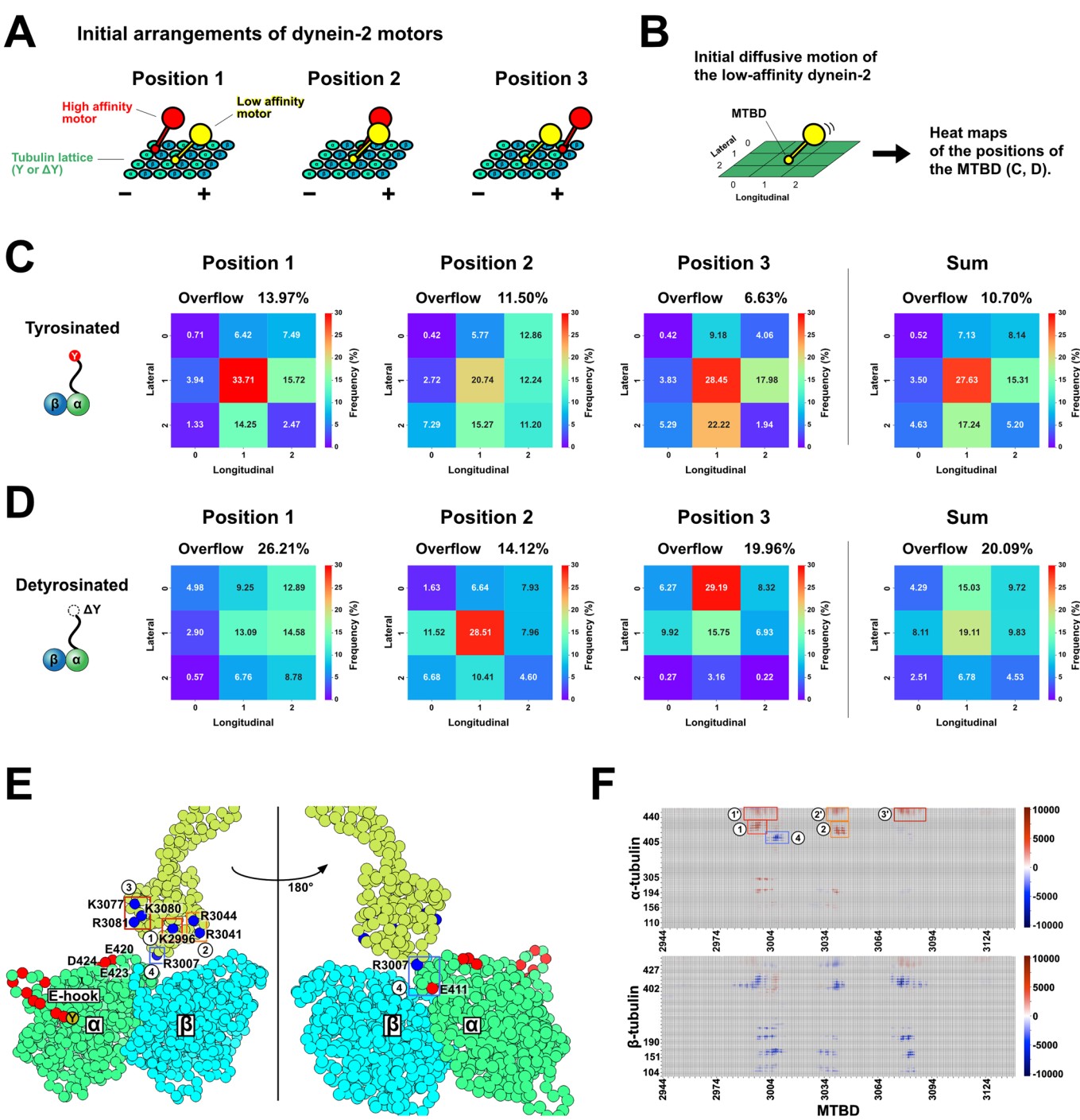

negatively charged glutamate residues in the E-hook become more exposed and form transient interactions with positively charged residues in MTBD, such as K3077, K3080, and R3081 (corresponding to positions ①′–③′ of Fig. 5F). In addition, contacts involving α-tubulin residues E420 and E423 become more frequent in ΔY, particularly at sites ① and ②, corresponding to K2996, R3041 and R3044 in MTBD. In contrast, the strong interaction between R3007 and α-tubulin E411 (contact ④) observed in the tyrosinated condition is diminished in ΔY, potentially leading to reduced MT-binding stability (Fig. 5F). These altered interactions likely underlie the distinct behaviors of dynein-2 on tyrosinated and detyrosinated tubulin lattices.

## Discussion

Here, we showed that dynein-2 exhibits higher affinity for ciliary doublets than dynein-1 by microtubule co-pelleting assay using doublets. Our cryo-ET analysis showed that dynein-2 preferentially binds to the A-tubule of the doublet, and our MD simulations

Figure 5.    MD simulation results of dynein-2 dimer on tyrosinated or detyrosinated tubulin lattice.

(A) Illustrations showing the setup of MD simulations. The low-affinity dynein-2 motor domain (yellow) and the high-affinity dynein-2 motor domain (red) were placed on the neighboring PF with three arrangements (positions 1, 2, and 3). Either tyrosinated (Y) or detyrosinated (ΔY) tubulin lattice was used. MT polarities are indicated. (B) A schematic of the trajectory analysis of the low-affinity motor. The initial diffusional motion of the low-affinity dynein-2 motor was simulated, and the positions of MTBD were plotted in a 3 × 3 area in (C, D). (C, D) Heatmaps of the low-affinity dynein-2 MTBD position after the MD simulations with tyrosinated tubulin in (B) and detyrosinated tubulin in (C). The mass center-of-gravity coordinates of the MTBDs of the low-affinity dynein-2 obtained from their trajectories are shown in the heat map (red: higher frequency; blue: lower frequency). The sum of the positions 1, 2, and 3 is shown in the rightmost panel. The center (1, 1) is the initial position of low-affinity dynein-2 MTBDs. Overflow refers to the cases where MTBDs diffused outside the 3 × 3 area, including detachment. See also Fig. EV5. (E) The snapshot of low-affinity MTBD highlighting contact sites ①–④. Positively charged residues in the MTBD are highlighted in blue, and negatively charged residues in the C-terminal E-hook of α-tubulin in red. (F) A difference map comparing contact maps of tyrosinated and detyrosinated conditions showing increases (red) and decreases (blue) in contacts upon detyrosination. In the detyrosinated state, E-hook interacted more intensively with MTBD, especially with the regions ①′, ②′, and ③′. Furthermore, the residues around E420 interacted with MTBD more frequently with MTBD regions ① and ② in the detyrosinated state. The contact site ④, which interacts more strongly in the tyrosinated condition, is essential for the stable binding of MTBD and tubulin. Source data are available online for this figure.

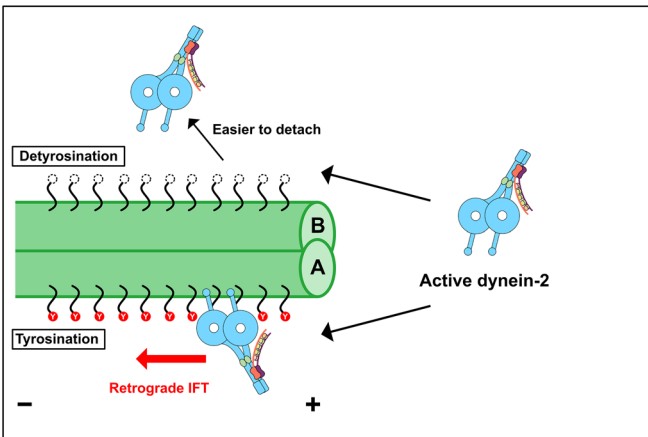

**Figure 6.    Model of the track recognition mechanism of dynein-2.**

At the tip of the cilia, dynein-2 will be released from anterograde IFT driven by kinesin-2. Released dynein-2 molecules get activated in unelucidated mechanisms by opening two motor domains. Activated open dynein-2 molecules bind to the doublets on either the A- or B-tubule side. Since the tubulin lattice from the B-tubule is detyrosinated, dynein-2 molecules are easier to detach. Dynein-2 molecules bound to the A-tubule tend to stay on the tyrosinated tubulin lattice, and these dynein-2 molecules exhibit unidirectional motion and power retrograde IFT.

indicated that dynein-2 has reduced affinity for the detyrosinated B-tubule.

Previous study of in situ IFT trains has shown that anterograde IFT and retrograde IFT trains utilize different tracks to avoid collision (Stepanek and Pigino, 2016). Our cryo-ET analysis of an in vitro reconstituted system from minimal components revealed that dynein-2 and doublet alone are sufficient for this track recognition. Unlike anterograde IFT trains, which span laterally around the B-tubule, retrograde IFT walks a narrow space of the A-tubule facing the membrane (Lacey et al, 2023; Lacey et al, 2024). Dynein-2's intrinsic preference for the A-tubule adapts to the spatial constraints of the track. In the anterograde IFT process, dynein-2 takes stacked conformation within IFT train cargo as transported by kinesin-2 (Jordan et al, 2018; Toropova et al, 2017). This configuration is important to keep the MTBDs of dynein-2 from interacting with doublets. At the ciliary tip, dynein-2 is released from the anterograde IFT train and activated by opening two heads. Based on our results, these active dynein-2 molecules can bind to doublets on either A- or B-tubule, but the dynein-2

molecules will be easily detached from the B-tubule side, and dynein-2 molecules on the A-tubule side start the retrograde IFT (Fig. 6).

In the previous study using *Chlamydomonas*, the A-tubule was shown to be enriched with tyrosinated tubulins and the B-tubule was enriched with detyrosinated tubulins by immunogold labeling (Johnson, 1998). Based on our MS analysis, tyrosinated α-tubulin was 33.7% and detyrosinated α-tubulin was 66.3%, roughly corresponding to 8 and 15 PFs, respectively (Fig. EV3). Although the exact timing of detyrosination of the doublet remains unclear, detyrosination likely occurs after the polymerization of the tubulins (Arce and Barra, 1985; Thompson et al, 1979). Since the C-termini of α-tubulins of PFs A1 and A11-13 are buried, these PFs should remain tyrosinated. PFs around A8/A9 that are the pathway of the retrograde IFT might also retain tyrosination (Fig. EV3F). This model will be tested in future studies using cryo-ET combined with labeling with tyrosination-state-specific antibodies.

Recently, it was shown that affinities of IFT trains to the A- or B-tubules were affected by the level of tyrosination/detyrosination of the doublets (Chhatre et al, 2025). Still, the molecular mechanism underlying this was not revealed. Our MD simulations suggest that dynein-2 recognizes the doublet's tyrosination/detyrosination states for the retrograde IFT trains. Dynein-2 MTBD has a more neutral or even slightly negative surface potential at the MT-binding interface, whereas dynein-1 has a prominent positively charged patch in the corresponding region (Fig. 2B), most likely for selecting its pathway within the doublets. Previously, H3 of dynein-1 MTBD was shown to play a role in the interaction with MT (Redwine et al, 2012). The absence of a positively charged residue in the H3 region of dynein-2 might contribute to its preference for tyrosinated tubulin because the glutamic acid residue is exposed at the C-terminus of the detyrosinated tubulin.

Our MD simulations indicate that dynein-2 prefers the tyrosinated tubulin lattice over the detyrosinated tubulin lattice. In our MD simulations, dynein-2 molecules tend to dissociate more readily from the detyrosinated tubulin lattice due to the altered interaction of the C-terminal tail of the detyrosinated α-tubulin with MTBD. Our contact map analysis in MD simulations identified several regions of the MTBD that are potentially involved in recognizing the tyrosination/detyrosination state of the tubulin lattice (Fig. 5E,F). In the detyrosinated tubulin lattice, there were more unconventional interactions between the tubulin dimer and dynein-2 MTBD, especially with the residues from H1, H4 and H6

(Fig. 5F, red). At the same time, the canonical interaction between the tubulin and H1 of dynein-2 MTBD was reduced (Fig. 5F, blue). This further supports our conclusion that the MTBD of dynein-2 is tuned for the tyrosinated tubulin lattice. This point will be further tested by high-resolution structural analyses and mutational analyses in future studies.

It is noteworthy that Position 3, which showed the most apparent difference between tyrosinated and detyrosinated tubulin lattice in the MD simulations, corresponds to the configuration most frequently observed in our cryo-ET (Fig. 4B). For kinesin-2, which drives the anterograde IFT, previous research has shown that the detyrosinated tubulin lattice enhances both its processivity and velocity (Sirajuddin et al, 2014). This difference in the sensitivities to the tyrosination state of dynein-2 and kinesin-2 might be crucial for sorting the retrograde and anterograde IFT. The cytoplasmic dynein from yeast, which lacks cilia, is not affected by the tyrosination states of the MT (Sirajuddin et al, 2014). In vitro study has shown that kinesin-1 itself is not affected by tyrosination state (Sirajuddin et al, 2014). Therefore, in the species having cilia, motors involved in IFT and doublets have co-evolved so that the doublets provide specific trajectories for IFTs of opposing directions. Similar specialization of MTs by tyrosination/detyrosination states has also been observed in singlets from the cytoplasm. As for the cytoplasmic dynein-1, while mammalian dynein-1 itself is insensitive to the tyrosination states of the MT (McKenney et al, 2016), mammalian dynein-1 has evolved its binding partners like CLIP-170 and p150$^{Glued}$ of dynactin complex to sense the tyrosination state of the singlet (McKenney et al, 2016; Nirschl et al, 2016). Dynein-2, which does not have modulators like CLIP-170 or dynactin, has likely tuned its MTBD to bind to tyrosinated tubulin lattice through evolution. Several kinds of kinesins are known to be sensitive to tyrosination/detyrosination states of singlets. For instance, kinesin-3 and kinesin-5 preferentially walk on tyrosinated singlets (Kahn et al, 2015; Tas et al, 2017). Based on these facts, tubulin tyrosination/detyrosination is a conserved strategy for sorting the motors between singlets and doublets.

There is, however, still a possibility that structural differences between the A- and B-tubules also contribute to the selective binding of dynein-2. We have previously shown that the doublet exhibits a deformation in the angles between PFs, particularly in the A-tubule (Ichikawa et al, 2019; Ichikawa et al, 2017). More recent cryo-EM studies confirmed that similar PF angle distortions are conserved across species (Zhou et al, 2023). In other types of dynein, like dynein-c (DHC9), flap structures are inserted between H2 and H3 of the MTBD (Kato et al, 2014). For γHC (ODA2) of outer arm dynein, LC1 is associated with the MTBD region (Ichikawa et al, 2015; Toda et al, 2020). These flap structures and LC1 are thought to be important for the interaction with the doublet area where two PFs have wider spaces (Lacey et al, 2019; Rao et al, 2021). Although dynein-2 does not have a flap or accessory proteins on MTBD, its dimeric structure might be favorable for binding to the specific PF geometry. It is also possible that both PF angles and tyrosination state cooperatively recruit dynein-2 to the specific area of the A-tubule.

Our results revealed that the MTBD of dynein-2 is fine-tuned for binding to the A-tubule. There are, however, still questions which have not been answered regarding the regulatory mechanisms of IFT. For instance, the mechanism by which dynein-2 is activated after being released from anterograde IFT remains unclear. Furthermore, while it is known that PTMs exhibit distinct patterns on doublets, the processes that generate these patterns remain to be elucidated.

In conclusion, we identified dynein-2 as a key component of the track recognition mechanism underlying retrograde IFT, an essential process for ciliary assembly, and one associated with human ciliopathies.

# Methods

**Reagents and tools table**

| Reagent/resource | Reference or source | Identifier or catalog number |
|---|---|---|
| **Experimental models** | | |
| *E. coli* (BL21(DE3)) | Lab stock | N/A |
| *E. coli* (DH10EMBacY) | Geneva Biotech | Cat #0501-DH10EMBacY |
| *E. coli* (DH10EMBac) | Thermo Fisher Scientific | Cat #10361012 |
| Sf9 | Thermo Fisher Scientific | Cat #11496015 |
| *Tetrahymena thermophila* | *Tetrahymena* Stock Center | SB255 |
| *Chlamydomonas reinhardtii* (WT) | *Chlamydomonas* Resource Center | CC-5325 mt- |
| *Chlamydomonas reinhardtii* (*VashL* mutant) | *Chlamydomonas* Resource Center | CLiP ID: LMJ.RY0402.233724 |
| **Recombinant DNA** | | |
| pET-42a vector containing sequence encoding SRS and dynein-1 MTBD with C-terminal 8 x His-tag. | This study | N/A |
| pET-42a vector containing sequence encoding SRS and dynein-2 MTBD with C-terminal 8 x His-tag. | This study | N/A |
| pFastBac vector containing sequences encoding ZZ-tag, TEV site, SNAPf-tag, GST-tag, and dynein-2 HC sequence (1,091-4,307) with mutations D1406A/Q1407A/R1410A | Toropova et al, 2017 | N/A |
| **Antibodies** | | |
| N/A | N/A | N/A |
| **Oligonucleotides and other sequence-based reagents** | | |
| N/A | N/A | N/A |
| **Chemicals, enzymes, and other reagents** | | |
| LB Broth | Difco | Cat #214906 |
| kanamycin | Sangon Biotech | Cat #A506636 |
| IPTG | Sangon Biotech | Cat #A600168 |
| Tris | Sigma-Aldrich | Cat #T6791 |
| NaCl | Diamond | Cat #A100241 |
| imidazole | Sangon Biotech | Cat #A600277 |
| PMSF | Sangon Biotech | Cat #A610425 |

| Reagent/resource | Reference or source | Identifier or catalog number |
|---|---|---|
| cOmplete™ EDTA-free Protease Inhibitor Cocktail | Roche | Cat #11873580001 |
| Ni-NTA agarose | QIAGEN | Cat #30250 |
| Insect-XPRESS™ Protein-free Insect Cell Medium with L-Glutamine | Lonza | Cat #12-730Q |
| PBS | Thermo Fisher Scientific | Cat #70011036 |
| SIM SF Expression Medium | Sino Biological, Inc. | Cat #MSF1 |
| Sf-900 II SFM (1x) | gibco | Cat #10902 |
| TBS | Lab stock | N/A |
| HEPES | Sigma-Aldrich | Cat #H3375 |
| KCl | Sigma-Aldrich | Cat #P5405 |
| TAP medium | Shanghai Guangyu Biological Technology Co., Ltd. | Cat #GY-M30-16 |
| ampicillin | Sangon Biotech | Cat #A610028 |
| potassium acetate | Sigma-Aldrich | Cat #791733 |
| magnesium acetate | Sigma-Aldrich | Cat #228648 |
| EGTA | Sigma-Aldrich | Cat #E4378 |
| glycerol | Sigma-Aldrich | Cat #G6279 |
| DTT | Sigma-Aldrich | Cat #D0632 |
| ATP | Sigma-Aldrich | Cat #A2383 |
| IgG Sepharose 6 resin | Cytiva | Cat #17096901 |
| TEV protease | Lab stock | N/A |
| GST· Bind™ resin | Millipore | Cat #70541 |
| Glutathione reduced | Sangon Biotech | Cat #A100399 |
| Proteose peptone No. 3 | Thermo Fisher Scientific | Cat #211693 |
| Glucose | Sangon Biotech | Cat #A501991 |
| Yeast extract | Thermo Fisher Scientific | Cat #211929 |
| Fe-EDTA | Sigma-Aldrich | Cat #EDFS |
| Dibucaine | Sigma-Aldrich | Cat #D0638 |
| MgSO$_4$ | Sinopharm Chemical Reagent Co., Ltd. | Cat #10013018 |
| Trehalose | Sigma-Aldrich | Cat #T5251 |
| Nonidet P-40 Alternative | Millipore | Cat #492016 |
| EDTA | Sigma-Aldrich | Cat #ED |
| PIPES | Sigma-Aldrich | Cat #P6757 |
| MgCl$_2$ | Sinopharm Chemical Reagent Co., Ltd. | Cat #10012818 |
| BSA | Sigma-Aldrich | Cat #B2064 |
| Gold nanoparticles (used for cryo-ET of in vitro reconstituted sample) | Sigma-Aldrich | Cat #752584 |

| Reagent/resource | Reference or source | Identifier or catalog number |
|---|---|---|
| KOH | Sinopharm Chemical Reagent Co., Ltd. | Cat #10017018 |
| Sucrose | Sangon Biotech | Cat #A100335 |
| Glacial acetic acid | Sinopharm Chemical Reagent Co., Ltd. | Cat #10000218 |
| PEG 20,000 | Sigma-Aldrich | Cat #8.18897 |
| Glutaraldehyde | Sinopharm Chemical Reagent Co., Ltd. | Cat #30092436 |
| HCl | Sinopharm Chemical Reagent Co., Ltd. | Cat #10011018 |
| 10-nm BSA tracer (used for cryo-ET of intact flagella) | Aurion | Cat #210.033 |
| Paclitaxel | Sangon Biotech | Cat #A601183 |
| Elastase | Sigma-Aldrich | Cat #E0127 |
| Trypsin inhibitor | Sigma-Aldrich | Cat #T9003 |
| ADP | Sigma-Aldrich | Cat #A2754 |
| Uranyl acetate | Electron Microscopy Sciences | Cat #22400 |
| Coomassie brilliant blue R-250 | Sangon Biotech | Cat #A610037 |
| IAA | Sigma-Aldrich | Cat #V900335 |
| Trypsin | Beijing Shengxia Proteins Scientific Ltd. | Cat #HLS TRY001C |
| FA | Thermo Fisher Scientific | Cat #A11750 |
| ACN | Thermo Fisher Scientific | Cat #A9554 |
| **Software** | | |
| ImageJ | Schneider et al, 2012 | https://imagej.net/ij/index.html |
| GraphPad Prism | GraphPad Software Inc. | https://www.graphpad.com/ |
| IMOD | Kremer et al, 1996 | https://bio3d.colorado.edu/imod/ |
| UCSF Chimera X | Meng et al, 2023 | https://www.cgl.ucsf.edu/chimerax/ |
| Tomography 5 | Thermo Fisher Scientific | N/A |
| Proteome Discoverer | Thermo Fisher Scientific | N/A |
| Mascot | Matrix Science | https://www.matrixscience.com/ |
| MODELLER | Webb and Sali, 2016 | https://salilab.org/modeller/ |
| COOT | Emsley and Cowtan, 2004 | https://www2.mrc-lmb.cam.ac.uk/personal/pemsley/coot/ |

| Reagent/resource | Reference or source | Identifier or catalog number |
|---|---|---|
| CafeMol | Kenzaki et al, 2011 | https://www.cafemol.org/ |
| **Other** | | |
| SORVALL LYNX 6000 Centrifuge | Thermo Fisher Scientific | Cat #75006590 |
| A27-8×50 rotor | Thermo Fisher Scientific | Cat #75003008 |
| NAP™-5 Columns | Cytiva | Cat #17-853-02 |
| FuGene HD transfection reagent | Promega | Cat #E2311 |
| Type 70 Ti rotor | Beckman Coulter | Cat #337922 |
| Amicon Ultra Centrifugal Filter Unit 100 kDa cutoff | Millipore | Cat #UFC810096 |
| TLA 110 rotor | Beckman Coulter | Cat #366735 |
| Nunc™ Cell-Culture Treated Multidishes 6 | Thermo Fisher Scientific | Cat #140675 |
| S55A2 rotor | himac | Cat #5720411009 |
| 4-15% precast gel for SDS-PAGE | Bio-Rad | Cat #4561086 |
| Quantifoil R2/2 200 mesh | Quantifoil | Cat #M2953C-1-200 |
| Quantifoil R2/1 300 mesh | Quantifoil | Cat #M2951C-1-300 |
| Vitrobot Mark IV | Thermo Fisher Scientific | N/A |
| Titan Krios | Thermo Fisher Scientific | N/A |
| Falcon II direct electron detector | Thermo Fisher Scientific | N/A |
| Phase plate | Thermo Fisher Scientific | N/A |
| Tecnai F20 microscope | Thermo Fisher Scientific | N/A |
| Ultrascan 4000 4k × 4k CCD camera | Gatan, Inc. | Cat #895 |
| Krios G4 Cryo-TEM | Thermo Fisher Scientific | N/A |
| Selectris X energy filter | Thermo Fisher Scientific | N/A |
| Falcon 4i detector | Thermo Fisher Scientific | N/A |
| carbon-coated grids | EM Japan | Cat #U1013 |
| Talos L120C TEM | Thermo Fisher Scientific | N/A |
| nanoflow EASYnLC 1200 system | Thermo Fisher Scientific | N/A |
| Orbitrap Exploris 480 mass spectrometer | Thermo Fisher Scientific | N/A |
| Orbitrap Fusion Lumos | Thermo Fisher Scientific | N/A |
| C18 analytical column | Dr. Maisch HPLC GmbH | Cat #MA-r119.aq.tc250.075 |

## Purification of MTBD constructs

The sequences coding human dynein-1 MTBD and part of coiled-coil region (NCBI Reference Sequence: NP_001367.2, 3266-3429 aa) or human dynein-2 MTBD and part of the coiled-coil region (NCBI Reference Sequence: NP_001368.2, 2962-3126 aa) were inserted into the modified pET-42a vector, respectively. The plasmids were transformed into $E.\ coli$ BL21(DE3) cells and the cells were cultured at 37 °C in LB media containing 30 µg/ml kanamycin until the $OD_{600}$ reached around 0.6. Protein expression was induced by adding 0.4 mM isopropyl-1-thio-β-D-galactopyranoside (IPTG) and incubated at 20 °C for 3 h. From here, the procedure was performed at 4 °C or on ice. Cells were harvested by centrifugation and were resuspended in lysis buffer (50 mM Tris-HCl pH 8.0, 150 mM NaCl, 20 mM imidazole) supplemented with 1 mM phenylmethylsulfonyl fluoride (PMSF) and cOmplete™ EDTA-free Protease Inhibitor Cocktail (Roche). The cells were homogenized, and the lysate was centrifuged at 15,000 g for 30 min using a SORVALL LYNX 6000 Centrifuge (Thermo Fisher Scientific) and an A27-8×50 rotor (Thermo Fisher Scientific). The supernatant was retrieved and incubated with Ni-NTA agarose (30250, QIAGEN) with rotation for 1 h. The column was washed with wash buffer (50 mM Tris-HCl pH 8.0, 250 mM NaCl, 20 mM imidazole), and bound proteins were eluted with elution buffer (50 mM Tris-HCl pH 8.0, 150 mM NaCl, 300 mM imidazole). The buffer of the elution fraction was exchanged for NAP5 buffer (50 mM Tris-HCl pH 8.0, 100 mM NaCl) by NAP™-5 Columns (17-853-02). The obtained proteins were frozen with liquid nitrogen and stored at −80 °C.

## Purification of GST-Dyn2 without ZZ-tag

GST-Dyn2 construct with mutations (D1406A/Q1407A/R1410A) was expressed and purified as in (Toropova et al, 2017). pFastBac vector containing ZZ-tag, TEV cleavage site, SNAPf-tag, GST tag, and dynein-2 HC sequence (1091–4307 aa, codon-optimized) between Tn7 sites was transformed to DH10EMBacY cells. Colonies with EMBacY bacmid containing GST-Dyn2 cassette between Tn7 sites were chosen by blue/white selection, cultured, and bacmids were purified. Sf9 cells (Thermo Fisher Scientific) were cultured in Insect-XPRESS Medium supplemented with L-glutamine at 27 °C with shaking at 100 rpm. Sf9 cells were transfected with bacmids using FuGene HD transfection reagent (Promega), and transfection efficiency was checked by YFP fluorescence from EMBacY. The media (V0) was collected and added to a 50 ml Sf9 culture. After incubation for 3 days, the cells were removed from the media by centrifugation, and the media (V1) was collected and stored at 4 °C. 2.5 ml of V1 was then added to 250 ml Sf9 culture. After three days, cells were harvested by centrifugation, washed with 1× PBS, flash-frozen in liquid nitrogen, and kept at −80 °C until purification. For purification, cells were resuspended in 20 ml purification buffer (30 mM 4-(2-hydroxyethyl)-1-piperazineethanesulfonic acid [HEPES] pH 7.4, 300 mM KCl, 50 mM potassium acetate, 2 mM magnesium acetate, 1 mM ethylene glycol tetraacetic acid [EGTA], 10% glycerol, 1 mM dithiothreitol [DTT], 0.2 mM adenosine triphosphate [ATP], 1 mM PMSF) containing cOmplete™ EDTA-free Protease Inhibitor Cocktail (Roche). Cells were lysed and ultracentrifuged (Type 70 Ti rotor, 183,960 g, 30 min, 4 °C), and the supernatant was incubated

with 1 ml of IgG Sepharose 6 resin (Cytiva) pre-equilibrated with purification buffer. The resin was collected by centrifugation (670 g, 5 min, 4 °C) and washed with 20 ml of purification buffer twice and 20 ml of TEV buffer (50 mM Tris pH 7.5, 150 mM potassium acetate, 2 mM magnesium acetate, 1 mM EGTA, 10% glycerol, 1 mM DTT, 0.2 mM ATP). The dynein-2 construct with SNAPf-tag and GST tag was removed from the resin by incubating with 4 ml of TEV buffer containing 100 µg TEV protease overnight at 4 °C with rotating. The eluted dynein-2 construct was concentrated using Amicon Ultra Centrifugal Filter Unit (100 kDa cutoff, Millipore), and ultracentrifuged (TLA 110 rotor, 337,932 g, 6 min), and the supernatant was flash-frozen in liquid nitrogen and stored in −80 °C. This construct was used for the cryo-ET analysis of in vitro reconstituted sample.

## Purification of GST-Dyn2 with ZZ-tag

pFastBac vector containing ZZ-tag, TEV cleavage site, SNAPf-tag, GST tag, and dynein-2 HC sequence (1091–4307 aa, codon-optimized) was transformed to DH10EMBac cells. Colonies with EMBac bacmid containing GST-Dyn2 cassette were selected by blue/white selection, cultured, and bacmids were purified. Sf9 cells (Thermo Fisher Scientific) were cultured in SIM SF Expression Medium (Sino Biological) at 27 °C with shaking at 130 rpm. Cells were transferred to the 6-well plate (Thermo Fisher Scientific), and the media was changed to Sf-900 II SFM. Then the Sf9 cells were transfected with bacmids using FuGene HD transfection reagent (Promega). After 5 days, the media (V0) was collected, and 500 µl of V0 was added to Sf9 cells in 50 ml Sf-900 II SFM. After incubation for 5 days, the cells were removed from the media by centrifugation, and the media (V1) was collected and stored at 4 °C. In total, 5 ml of V1 was used to infect 500 ml Sf9 culture in SIM SF Expression Medium (Sino Biological). After 16 h at 27 °C with shaking at 130 rpm, the temperature was changed to 20 °C, and incubated for 72 h. Cells were then harvested by centrifugation, washed with 1× TBS, flash-frozen in liquid nitrogen, and kept at −80 °C until purification. For purification, cells were resuspended in 20 ml purification buffer containing cOmplete™ EDTA-free Protease Inhibitor Cocktail (Roche). Cells were lysed and ultracentrifuged (Type 45 Ti rotor, 40,000 rpm, 30 min, 4 °C), and the supernatant was incubated with 1 ml of GST· Bind™ resin (Millipore) pre-equilibrated with TEV buffer. The resin was collected by centrifugation and washed with 20 ml of TEV buffer three times. The dynein-2 construct with ZZ-tag, SNAPf-tag, and GST tag was eluted from the resin by incubating with TEV buffer containing 10 mM reduced glutathione. The buffer of eluted dynein-2 construct was exchanged to TEV buffer without glutathione, and concentrated using Amicon Ultra Centrifugal Filter Unit (100 kDa cutoff, Millipore). The proteins were flash-frozen in liquid nitrogen and stored in -80 °C. This construct was used for the binding experiments of the *Chlamydomonas* doublet.

## Purification of doublets with clean outer surface from *Tetrahymena* cilia

Native doublets with clean outer surfaces were prepared from *Tetrahymena thermophila* as described in (Black et al, 2021). *Tetrahymena* cells (SB255 strain) were cultured in 1 L of SPP media (1% proteose peptone No. 3, 0.2% glucose, 0.1% yeast extract, 0.003%

ethylenediaminetetraacetic acid iron (III) sodium salt [Fe-EDTA]) until the $OD_{600}$ reached 0.7. Cells were harvested by low-speed centrifuge (700× g, 10 min, 4 °C) and concentrated to 25 ml with SPP media. The procedure was performed at 4 °C hereafter otherwise noted. Cilia were removed from the cell bodies by adding the final 1 mg/ml dibucaine and swirling for 1 min. The cell bodies were removed by slow-speed centrifugation (2000× g, 7 min, 4 °C), and the cilia were pelleted with higher-speed centrifugation (17,000× g, 30 min, 4 °C). Cilia pellet was then resuspended in cilia final buffer (CFB) (50 mM HEPES pH 7.4, 3 mM MgSO4, 0.1 mM EGTA, 0.5% Trehalose, 1 mM DTT) supplemented with 1 mM PMSF. Sequential purification was performed to obtain doublets without protein structures bound outside. First, cilia membranes were removed from the axonemes by adding the final 1.5% Nonidet P-40 Alternative (Millipore, 492016) for 30 min. Then, the demembraned axoneme was spun down by centrifuge (7800× g, 10 min, 4 °C). The pellet was resuspended with CFB supplemented with 1 mM PMSF and 0.4 mM ATP and incubated at room temperature for 5 min to release each doublet from the axoneme. After this, the final 0.6 M NaCl was added and incubated for 30 min on ice to deplete axonemal dyneins. The sample was centrifuged (16,000× g, 10 min, 4 °C) and resuspended in CFB containing 0.6 M NaCl to remove the remaining axonemal dyneins. Doublets were then dialyzed against low-salt buffer (5 mM HEPES pH 7.4, 1 mM DTT, 0.5 mM EDTA) at 4 °C overnight to remove radial spokes. The clean doublets after dialysis were centrifuged again (16,000× g, 10 min, 4 °C), and the pellet was finally resuspended in B80-TK buffer (80 mM piperazine-N,N′-bis(2-ethanesulfonic acid) [PIPES] pH 6.9, 2 mM MgCl2, 1 mM EGTA, 1 mM DTT, 20 µM paclitaxel, 50 mM KCl).

## MT co-pelleting assay

Dynein-1 and dynein-2 MTBD constructs were ultracentrifuged (45,000 rpm for 30 min at 4 °C, Himac S55A2 rotor), and the supernatant was used for the experiments. The concentrations of dynein-1 and dynein-2 MTBD constructs in the supernatant were determined based on the band intensities on SDS-PAGE gel using BSA as standards. 0.23 µM MTBD constructs were mixed with 120 µg/ml *Tetrahymena* doublets in NAP5 buffer, incubated on ice for 45 min, and ultracentrifuged (45,000 rpm for 30 min at 4 °C, himac S55A2 rotor). The supernatant and pellet fractions were separated and analyzed by SDS-PAGE, and the band intensities were analyzed using ImageJ (NIH) (Schneider et al, 2012). Technical replicates were performed.

## In vitro reconstitution

Purified *Tetrahymena* doublets (final 500 µg/ml) were incubated with GST-Dyn2 (final 450 µg/ml) for 30 min at 25 °C and then centrifuged (16,000× g, 10 min, 4 °C). The supernatant was removed, and the precipitation fraction was resuspended in B80-TK buffer and used for vitrification after adding BSA-treated gold nanoparticles (10-nm diameter) prepared as in (Iancu et al, 2006) at an optimal ratio.

## Cryo-ET analysis of GST-Dyn2 decorated doublet

In all, 3.5 µl reconstituted sample was added to glow-discharged grids (Quantifoil R2/2) and vitrified using Vitrobot Mark IV (Thermo Fisher Scientific) with blot force 3 and blot time 3 s.

For dataset 1, the grids were loaded into Titan Krios (Thermo Fisher Scientific), operating at 300 kV, equipped with Falcon II direct electron detector (Thermo Fisher Scientific). The tilt series were acquired from −40 to +60 degrees tilts with 4-degree increments (26 images) with a total dose of 79.7 electrons/Å². The nominal magnification was 29,000×, and the defocus was −0.5 or −1 μm. The phase plate's activation time was 70 s.

For dataset 2 (technical replicate), the grid was loaded into Tecnai F20 microscope (Thermo Fisher Scientific), operating at 200 kV, equipped with Gatan Ultrascan 4000 4k × 4k CCD camera. The tilt series were acquired from −60 to +60 degrees tilts with 3-degree increments (41 images), at a nominal magnification of 19,000×, with a total dose of 70 electrons/Å². A defocus of −8 μm was used to increase the contrast.

The tomograms were reconstructed with Etomo from IMOD (Kremer et al, 1996) using the Simultaneous Iterative Reconstruction Technique (SIRT) method with binning of 3.

### Chlamydomonas cell culture and flagella purification

*Chlamydomonas* cell culture and flagella purification were performed as in (Black et al, 2021). The cells were cultured in 1 L tris-acetate-phosphate (TAP) medium under a 12-hour light/dark cycle with 100 μg/ml ampicillin. The cultures were stirred continuously at 320 rpm using a magnetic stirrer for 5–7 days. For harvesting, cells were centrifuged at 700× g for 7 min at 4 °C. The resulting pellet was resuspended with 1 L of deionized water. The cells were subsequently incubated in the dark with stirring at 320 rpm for 1 h to promote flagella growth. Cells were centrifuged at 700× g for 10 min at 4 °C. The supernatant was discarded, and the pellet was resuspended with HMDS buffer (10 mM HEPES-KOH pH 7.4, 5 mM MgSO₄, 4% sucrose, 1 mM DTT). The pH was changed to 4.5 by adding glacial acetic acid and incubated for 1 min with stirring, then promptly neutralized to pH 7.5 with KOH. The sample was centrifuged at 1800× g for 5 min at 4 °C, and the supernatant was carefully collected. This was followed by centrifugation at 4000 rpm for 1 h at 4 °C. The resulting flagella pellet was resuspended in 500 μL HMDS buffer. The cell debris was removed by centrifuging at 480× g for 5 min at 4 °C, and the clean flagella were collected by centrifuging at 9200 rpm for 10 min at 4 °C. Flagella were snap-frozen in liquid nitrogen and stored at −80 °C for subsequent procedures.

### Cryo-ET analysis of intact Chlamydomonas flagella

Flagella from CC-125 (WT) strain were resuspended in 250 μL HMDEKP buffer (30 mM HEPES pH 7.4, 5 mM MgSO₄, 0.5 mM EGTA, 25 mM potassium acetate, 1 mM DTT, 0.5% polyethylene glycol, MW: 20,000 [PEG 20,000]). Crosslinking was performed with 0.15% glutaraldehyde for 30 min on ice. The reaction was quenched by adding Tris-HCl pH 7.5 to a final concentration of 35 mM. Samples were centrifuged at 9200 rpm for 10 min at 4 °C, and the pellet was resuspended with HMDEKP buffer so that the flagella are in the appropriate concentration, and mixed with 10-nm BSA tracer (Aurion, 210.033). Prior to application, grids (Quantifoil R2/1) were glow-discharged for 1 min at 20 mA to improve hydrophilicity. 4 μL sample was applied to each grid. Sample vitrification was performed using Vitrobot Mark IV (Thermo Fisher Scientific) at 23 °C and 100% humidity. Grids

were blotted for 8 s with blot force 0 with a wait time of 45 s. Cryo-ET data of intact flagella were collected by Krios G4 Cryo-TEM equipped with a Selectris X energy filter (slit width 10 eV) and Falcon 4i detector (Thermo Fisher Scientific) operating at 300 kV. A grouped dose-symmetric scheme from −60° to 60° was used with a 3° increment at a dose per tilt of 4 e⁻/Å² (total 164 e⁻/Å²). Tilt series were collected at 64,000x nominal magnification, corresponding to a pixel size of 1.96 Å, using Tomography 5 software (Thermo Fisher Scientific). Tomograms were then reconstructed using IMOD (Kremer et al, 1996) by the backprojection method with binning of 3.

### Purification of doublets with a clean outer surface from *Chlamydomonas* flagella

Flagella purified from CC-5325 (WT) and the *VashL* mutant strain (CLiP ID: LMJ.RY0402.233724) *Chlamydomonas* cells were resuspended in HMDEKP buffer containing 10 μM paclitaxel. First, the axonemes were obtained by incubating with 1.5% NP40 on ice for 30 min. Then, axonemes were collected by centrifugation at 7800× g for 10 min at 4 °C and resuspended with HMDEKP buffer with 10 μM paclitaxel. For splitting the doublets for EM, the axonemes were digested with 5 μg/mL elastase (E0127, Sigma-Aldrich) in the presence of 5 μg/mL trypsin inhibitor (T9003, Sigma-Aldrich) for 90 s at room temperature, and the reaction was quenched by adding 10 mM PMSF. This step was skipped for the sample preparation for the MS analysis. Then, the mixture was sequentially incubated with 1 mM adenosine diphosphate (ADP) for 10 min and 0.1 mM ATP for another 10 min at room temperature. Samples were then centrifuged at 16,000× g for 10 min at 4 °C and resuspended in HMDEKP buffer with 1 mM DTT, 10 μM paclitaxel and 1 mM PMSF. Split doublets were resuspended with HMDEKP buffer containing 0.6 M NaCl on ice for 30 min and centrifuged at 16,000× g for 10 min at 4 °C twice to remove associated proteins.

### MS analysis

Salt-treated *Chlamydomonas* doublet proteins were separated using one-dimensional SDS-PAGE gels and stained with Coomassie brilliant blue (CBB). Tubulin bands were excised for in-gel digestion. Proteins were reduced by 10 mM DTT for 30 min at 56 °C. The extracts were alkylated with 50 mM iodoacetamide (IAA) at room temperature for 45 min in the dark. Sample digestion was carried out using 5 ng/μl sequencing-grade modified trypsin overnight at 37 °C. 10% formic acid (FA) was used to quench digestion. Supernatants were collected, and the peptides were extracted with a 30% acetonitrile (ACN). The extracted tryptic peptides were dried under vacuum and resuspended for LC-MS/MS analysis.

LC-MS/MS analysis was performed using a nanoflow EASYnLC 1200 system (Thermo Fisher Scientific) coupled to an Orbitrap Exploris 480 mass spectrometer or Orbitrap Fusion Lumos (Thermo Fisher Scientific). A single-column setup was adopted for all analyses. Samples were analyzed on a home-packed C18 analytical column (75 μm inner diameter × 25 cm, ReproSil-Pur C18-AQ, 120 Å 1.9 μm (Dr. Maisch HPLC GmbH)) (Kovalchuk et al, 2019). The mobile phases consisted of Solution A (0.1% FA) and Solution B (0.1% FA in 80% ACN).

The MS/MS data were searched against UniProt *Chlamydomonas reinhardtii* protein database (18,828 entries, downloaded on May 19th, 2025) using Proteome Discoverer (version 2.4, Thermo Fisher Scientific) with Mascot (version 2.7.0, Matrix Science) as the search engine (Perkins et al, 1999). To distinguish between tyrosinated and detyrosinated forms of α-tubulin C-terminal peptides, the database search included both the canonical FASTA sequence of *Chlamydomonas* α-tubulin (UniProt ID: Q540H1) and a modified version lacking the C-terminal tyrosine residue. The mass tolerances were set for 10 ppm for precursor and 0.05 Da for fragment. Up to two missed cleavages were allowed. The carbamidomethylation on cysteine was set as a fixed modification, and acetylation at the protein N-terminus and oxidation on methionine were set as variable modifications. To evaluate glutamylation, this modification was added as a variable modification during the database search.

## Negative stain EM of *Chlamydomonas* doublets decorated with GST-Dyn2

Equal volumes of 300 μg/ml salt-treated *Chlamydomonas* doublets in HMDEKP buffer and 37.6 μg/ml GST-Dyn2 in TEV buffer were mixed and incubated on ice for 30 min in the presence of 10 μM paclitaxel. As a negative control, salt-treated *Chlamydomonas* doublets were mixed with TEV buffer and incubated under the same condition. Then, 3.5 μl of each sample was applied to the carbon-coated grids (U1013, EM Japan) that had been pre-hydrophilized by glow discharge, and stained with 1.5% uranyl acetate. The grids were loaded into a Talos L120C TEM (Thermo Fisher Scientific) operating at 120 kV, and the images were recorded at a nominal magnification of 57,000x.

## Model building for MD simulations

Model building was performed as described in (Kubo and Bui, 2023). In brief, the tubulin lattice model for MD simulations was generated by MODELLER (Webb and Sali, 2016) using TUBA1B (UniProtKB: P68363) and TUBB (UniProtKB: P07437) sequences and cryo-EM structure of GDP MT (PDB ID: 6DPV) as the reference structure. For the detyrosinated tubulin lattice, the tyrosine residue at the C-terminus of the α-tubulin was removed. For the dynein-2 dimer model, the high-affinity model of the dynein-2 motor domain and the low-affinity model of the dynein-2 motor domain were generated as in (Kubo and Bui, 2023). The DYHC2 (UniProtKB: Q8NCM8) sequence was used for model building. For the low-affinity structure, dynein-2 motor domain structure in ADP·Vi condition (PDB ID: 4RH7) (Schmidt et al, 2015) was used as a reference. High-affinity dynein-2 model was generated by combining AAA+ and stalk region structure (PDB ID: 3VKH) (Kon et al, 2012), stalk and MTBD structure (PDB ID: 3J1T) (Redwine et al, 2012), and MTBD-tubulin interface (PDB ID: 6KIQ) (Nishida et al, 2020). First, 3VKH and 3J1T structures were combined by MD simulations as in (Kubo et al, 2017). The model was combined with 6KIQ using COOT (Emsley and Cowtan, 2004). Finally, a homology model was generated using MODELLER (Webb and Sali, 2016).

## MD simulations

Four PFs, each containing three tubulin dimers, and two dynein-2 motor domains corresponding to one dynein-2 dimer, were used for the MD simulations. In each of the tyrosinated and detyrosinated tubulin lattices, the low-affinity dynein-2 head was placed at the center of the tubulin lattice ((1, 1) in Fig. 5B), and the high-affinity dynein-2 head was placed in the PF on the far side of the paper (Fig. 5A). The low-affinity dynein-2 structure was docked onto the tubulin lattice based on PDB ID: 3J1U (Redwine et al, 2012). High-affinity dynein-2 was placed in three positions relative to low-affinity dynein-2: backward, lateral, and forward (Fig. 5A). The high-affinity dynein-2 head domain's MTBD was set according to the MTBD position from PDB ID: 6KIQ (Nishida et al, 2020). Three distinct initial configurations of the high-affinity dynein-2 head were prepared: backward, lateral, and forward, each defined relative to the position of the low-affinity head (Fig. 5A).

MD simulations were performed using CafeMol version 2.1 (Kenzaki et al, 2011) as in (Kubo and Bui, 2023) for each simulation with $3 \times 10^7$ MD steps. The equations of motion were integrated using underdamped Langevin dynamics. For each configuration, 20 independent simulations were performed using different random seeds, resulting in a total of 60 simulations per tubulin modification state. The simulations employed an implicit solvent model with a relative dielectric constant of 78, an ionic strength of 0.1 M, and a temperature of 300 K. The friction coefficient was set to 2.0 (CafeMol unit), and default values were used for other parameters. For intra- and inter-molecules of the proteins, AICG2 + force field, electrostatic interaction, and excluded volume function were considered. Heatmaps (Fig. 5C,D) were generated based on the mass center-of-gravity coordinates of the MTBDs of the low-affinity dynein-2 head obtained from their trajectories. To reduce dependence on the initial configuration, the first $5 \times 10^6$ MD steps were excluded from the analysis.

## Contact map analysis

Contact maps were constructed by identifying residue pairs between the MTBD and α/β-tubulin that were within 1.0 nm across each simulation snapshot. The number of contacts for each residue pair was accumulated over the full trajectory time. A differential contact map was then computed by subtracting the Y contact frequencies from those in the ΔY condition, highlighting contact sites that were preferentially formed under one condition or the other.

## Statistical analysis

Statistical analyses were performed using GraphPad Prism 9.

## Visualization

Models and structures were visualized using ChimeraX (Meng et al, 2023).

# Data availability

The representative tomogram data from this publication have been deposited to the Electron Microscopy Data Bank (EMDB) database (https://www.ebi.ac.uk/pdbe/emdb) and assigned the identifier EMD-66326. Other datasets analyzed in this study are available from the corresponding author upon reasonable request.

The source data of this paper are collected in the following database record: biostudies:S-SCDT-10_1038-S44318-025-00648-1.

## Peer review information

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

## Acknowledgements

The authors appreciate Dr. Anthony J. Roberts from the University of Oxford for generously providing GST-Dyn2 proteins for initial experiments and MTBD constructs, as well as for his critical reading of the manuscript. We thank Dr. Khanh-Huy Bui from McGill University for his insightful advice. Cryo-ET data of in vitro reconstituted samples were collected with help from the Facility for Electron Microscopy Research at McGill University, especially from Dr. Kaustuv Basu. Cryo-ET data of intact flagella and negative-staining EM data were collected with help from Dr. Dejian Zhou and Dr. Jianjiang Mao from the EM Platform of State Key Laboratory of Genetics and Development of Complex Phenotypes, School of Life Sciences at Fudan University. We thank Ms. Lin Huang from the Proteomics Platform of State Key Laboratory of Genetics and Development of Complex Phenotypes, School of Life Sciences at Fudan University for her assistance with MS. We thank Prof. Motoyuki Hattori and Mr. Linghao Zhang from Fudan University for technical support. We are grateful to Prof. Yagi from the Prefectural University of Hiroshima for help in obtaining CC-125 *Chlamydomonas* flagella. We appreciate the reviewers for their time and effort in improving the manuscript. This work was supported by the Natural Science Foundation of Shanghai (24ZR1403800) and JST PRESTO (JPMJPR20E1) to MI, JSPS KAKENHI (22K15070) to SK, and the International Collaborative Research Program of Muroran Institute of Technology to AK.

## Author contributions

**Haoqiang K He**: Data curation; Formal analysis; Investigation; Writing—original draft. **Shintaroh Kubo**: Conceptualization; Data curation; Formal analysis; Funding acquisition; Investigation; Visualization; Writing—original draft; Writing—review and editing. **Xuwei Chen**: Data curation; Investigation; Visualization; Writing—original draft. **Qianru H Lv**: Data curation; Investigation; Writing—original draft. **Azusa Kage**: Data curation; Funding acquisition; Investigation; Writing—original draft; Writing—review and editing. **Muneyoshi Ichikawa**: Conceptualization; Data curation; Formal analysis; Supervision; Funding acquisition; Validation; Investigation; Visualization; Methodology; Writing—original draft; Project administration; Writing—review and editing.

Source data underlying figure panels in this paper may have individual authorship assigned. Where available, figure panel/source data authorship is listed in the following database record: biostudies:S-SCDT-10_1038-S44318-025-00648-1.

## Disclosure and competing interests statement

The authors declare no competing interests.

# Expanded View Figures

**Figure EV1. Sequence alignment of MTBD regions from dynein-2 and dynein-1 HCs.** ▶

Amino acid sequence of the MTBD region of human dynein-2 HC (NCBI Reference Sequence: NP_001368.2), *Mus musculus* dynein-2 HC (NCBI Reference Sequence: NP_084127.2), *Danio rerio* dynein-2 HC (NCBI Reference Sequence: NP_001410228), *Xenopus laevis* dynein-2 HC (NCBI Reference Sequence: XP_041438615), *Drosophila melanogaster* dynein-2 HC (NCBI Reference Sequence: NP_001036369), *Caenorhabditis elegans* dynein-2 HC (NCBI Reference Sequence: NP_492221.2), *Chlamydomonas reinhardtii* dynein-1b HC (NCBI Reference Sequence: XP_001696428.1), human dynein-1 HC (NCBI Reference Sequence: NP_001367.2), *Mus musculus* dynein-1 HC (NCBI Reference Sequence: NP_084514.2), *Xenopus laevis* dynein-1 HC (NCBI Reference Sequence: XP_018086051.1), *Danio rerio* dynein-1 HC (NCBI Reference Sequence: NP_001036210.1), *Drosophila melanogaster* dynein-1 HC (NCBI Reference Sequence: NP_001261430.1), *Caenorhabditis elegans* dynein-1 HC (NCBI Reference Sequence: NP_491363.1) were aligned using Clustal W (Thompson et al, 1994) and figure was prepared using ESPript 3.0 (Robert and Gouet, 2014). Note that *Chlamydomonas reinhardtii* dynein-1b is IFT dynein equivalent to dynein-2. Cyan highlights show the amino acid residues conserved only in dynein-2 and yellow highlights indicate the amino acid residues conserved only in dynein-1. White arrowheads indicate the residues conserved either in dynein-2 or dynein-1.

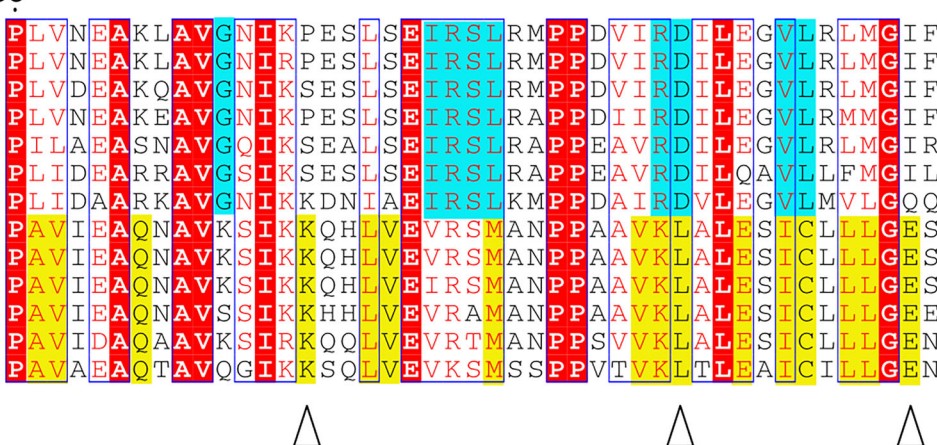

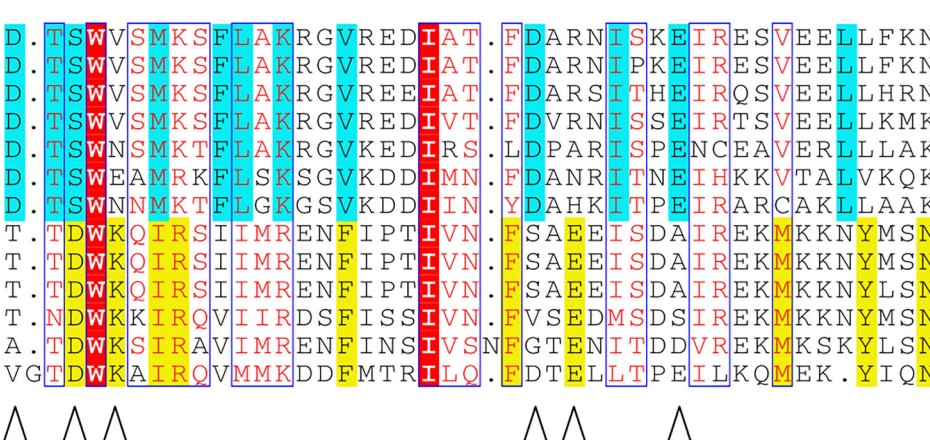

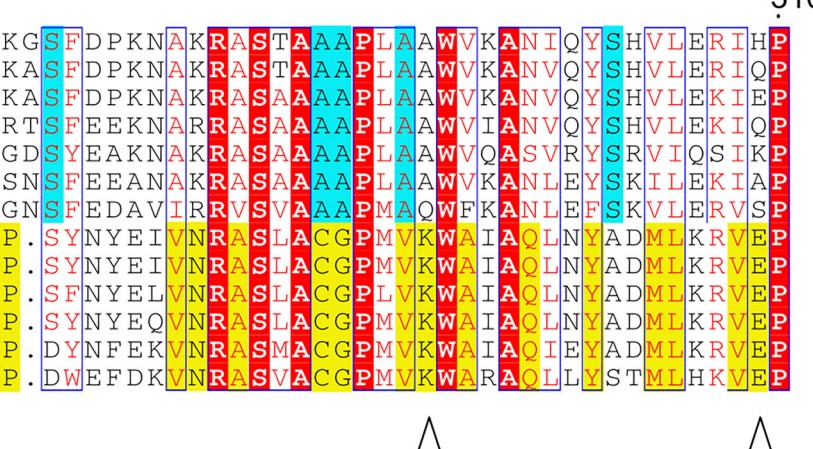

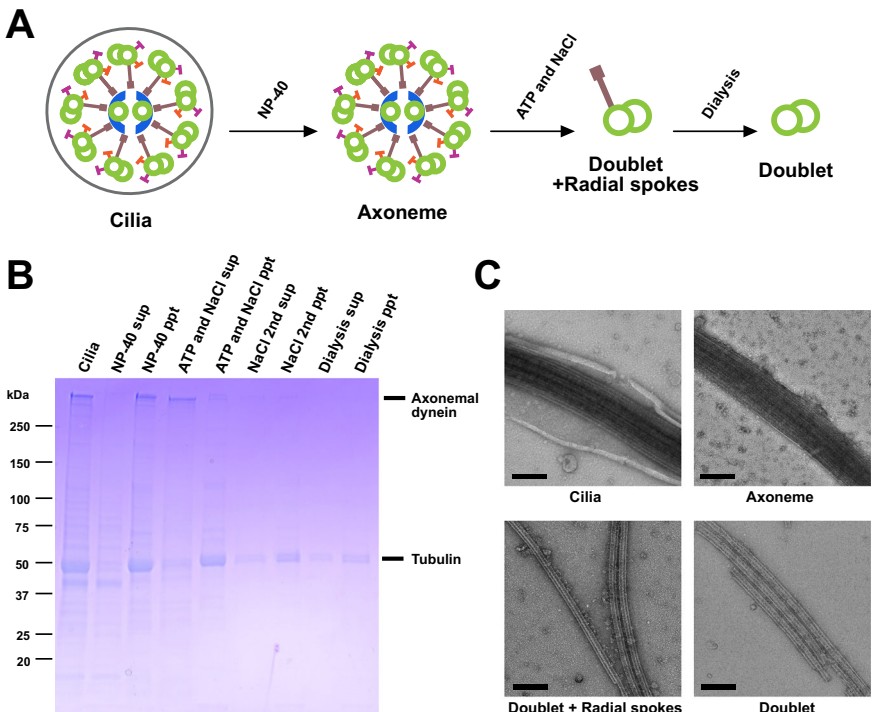

**Figure EV2. Preparation of *Tetrahymena* doublets with clean outer surface.**

(**A**) The workflow of doublet purification with a clean outer surface. (**B**) SDS-PAGE gel of sequential purification of *Tetrahymena* doublet. Associated proteins, especially axonemal dyneins, are removed while the tubulin band is visible. (**C**) Negative staining EM images of sequentially purified doublet samples. The doublets with clean outer surfaces were obtained. Scale bars: 200 nm.

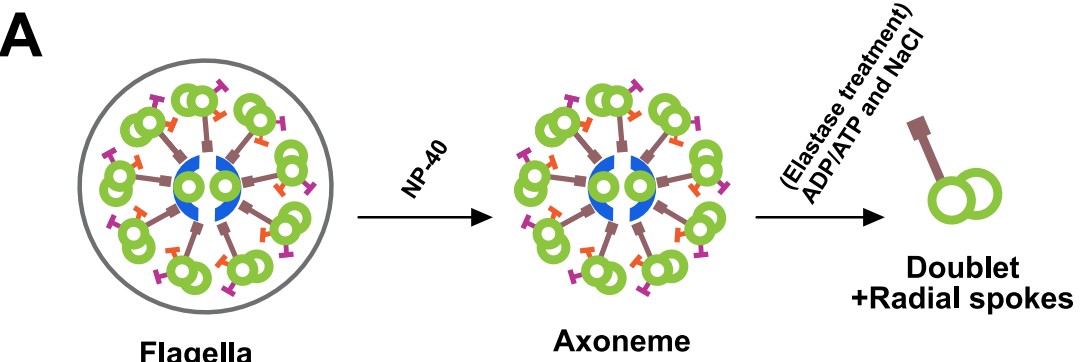

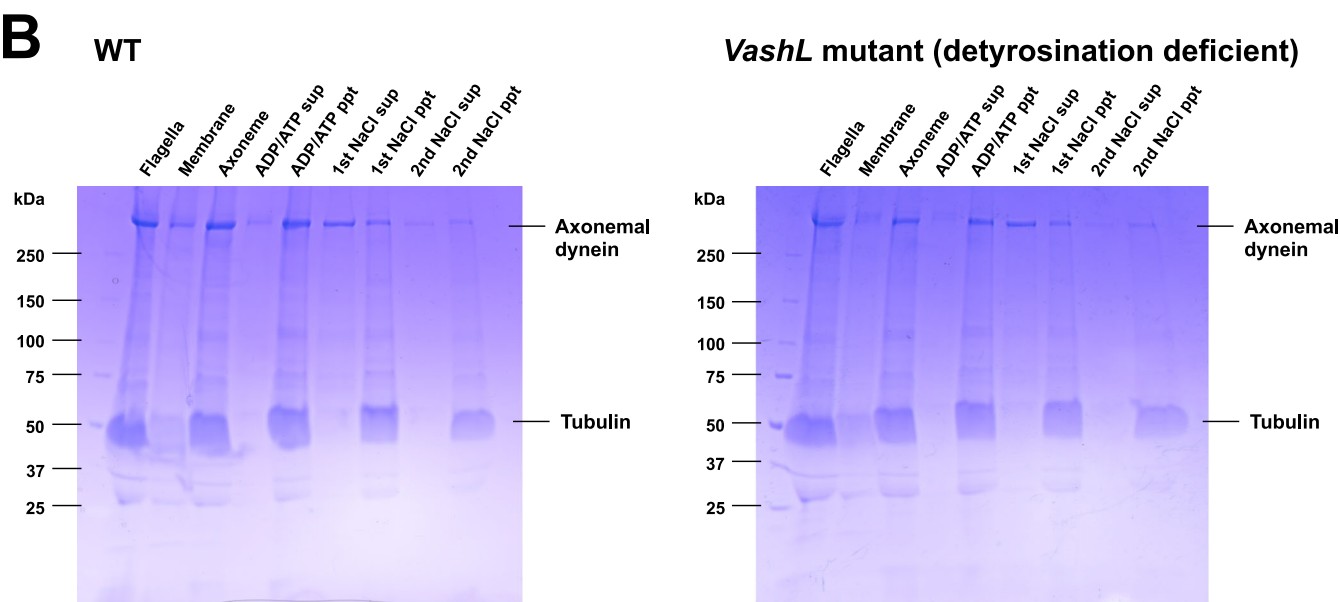

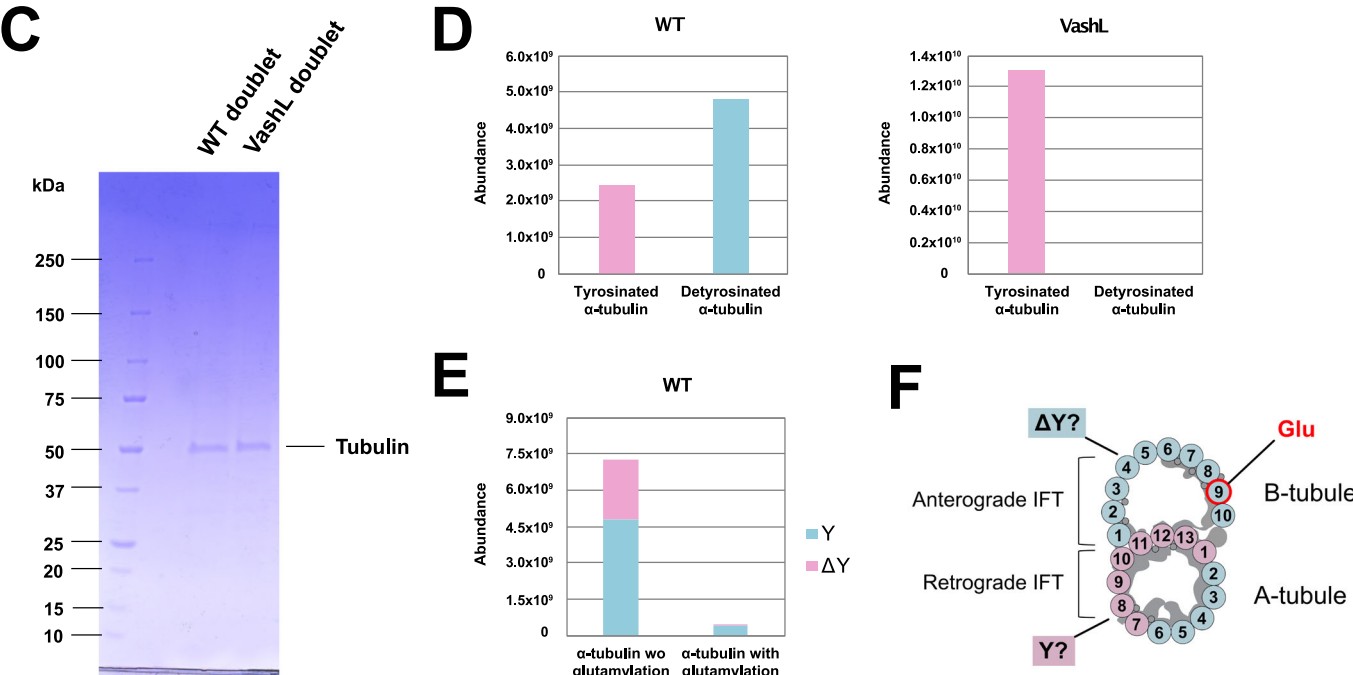

◀ **Figure EV3. Data related to *Chlamydomonas* doublets.**

(A) The workflow of doublet purification from *Chlamydomonas* flagella. The elastase treatment was performed when doublets were used for EM analysis. For doublets used for the MS analysis, elastase treatment was not performed. (B) SDS-PAGE gels of sequential purification of *Chlamydomonas* doublets from either WT or *VashL* mutant cells. (C) SDS-PAGE gel to compare doublets purified from WT and *VashL Chlamydomonas* flagella. (D) Detection of tyrosinated and detyrosinated C-terminal peptides of α-tubulin by MS. Bar graphs show the abundance of tyrosinated and detyrosinated α-tubulin C-terminal peptides detected in WT (left) and *VashL* (right) doublet samples. In the WT sample, abundance values of tyrosinated peptides was $2.45 \times 10^9$ and that of detyrosinated peptides was $4.81 \times 10^9$. In contrast, only tyrosinated α-tubulin peptides were detected in the *VashL* mutant sample with abundance value of $1.30 \times 10^9$. (E) Detection of C-terminal peptides of α-tubulin with and without glutamylation by MS. The MS result of WT was re-analyzed based on whether C-terminal peptides have glutamylation, and the abundance of each form was plotted. The abundance values were as follows: ΔY wo E, $4.81 \times 10^9$; Y wo E, $2.45 \times 10^9$; ΔY with E, $3.83 \times 10^8$; Y with E, $1.10 \times 10^8$. (F) Schematic of possible localization of tyrosinated and detyrosinated PFs. Retrograde and anterograde IFT tracks are indicated. The location of glutamylated PF is based on (Alvarez Viar et al, 2024).

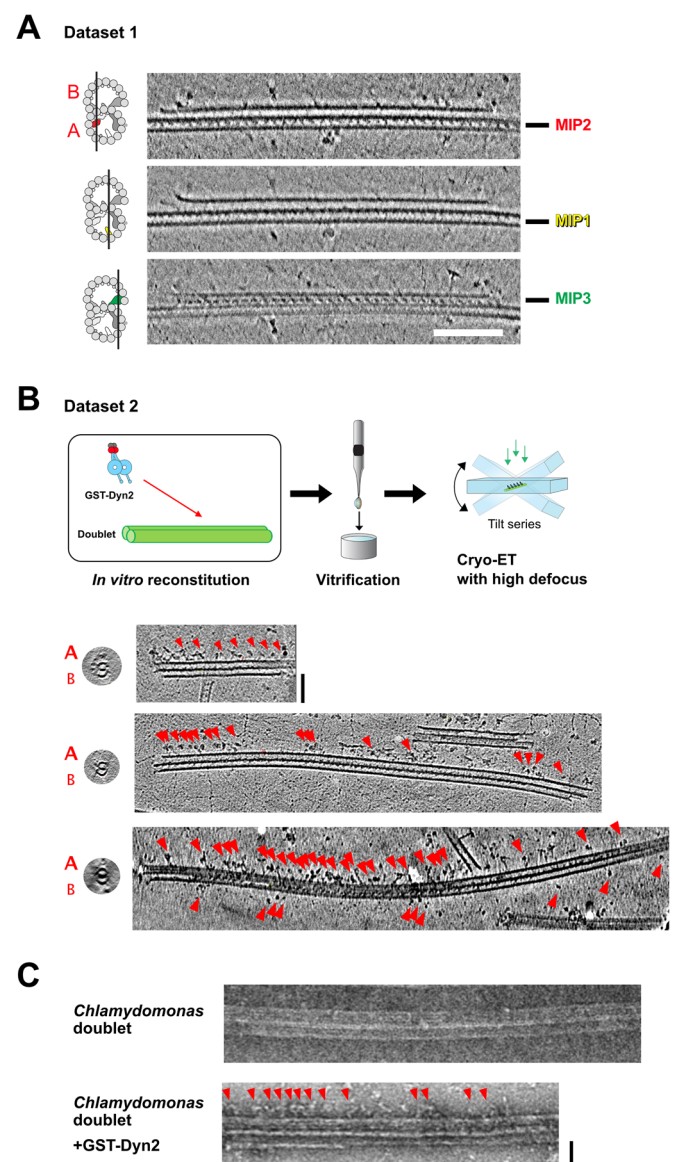

**Figure EV4.   Data related to decoration of doublet by GST-Dyn2.**

(**A**) Longitudinal tomographic slice of the representative reconstructed doublet's 3D structure. Black lines in the illustrations of the doublet on the left indicate the sections on the right panels. MIP structures distinctive for A- or B-tubules are highlighted in the model and indicated in the tomographic slices. Naming and colorings are adopted from (Ichikawa et al, 2017; Ichikawa et al, 2019). Scale bar, 100 nm. (**B**) Cryo-ET workflow for dataset 2 (top), and representative tomographic slices (bottom). Cross-sectional (left) and longitudinal (right) views of the doublets decorated with GST-Dyn2 molecules (red arrowheads). A- and B-tubules of the doublets are indicated, with the A-tubule oriented toward the top. More GST-Dyn2 molecules were observed on the A-tubule sides. Tilt series for this dataset were acquired with a high defocus (−8 µm) instead of using the VPP. Scale bar, 100 nm. (**C**) Results of decoration of *Chlamydomonas* doublets by GST-Dyn2 molecules. Typical negative stain EM images of salt-treated *Chlamydomonas* doublet without (top) and with (bottom) incubation with GST-Dyn2. GST-Dyn2 molecules (red arrowheads) were observed to accumulate along one side of the doublets, presumably corresponding to the A-tubule side, judged by the thickness of the tubules. Scale bar, 50 nm.

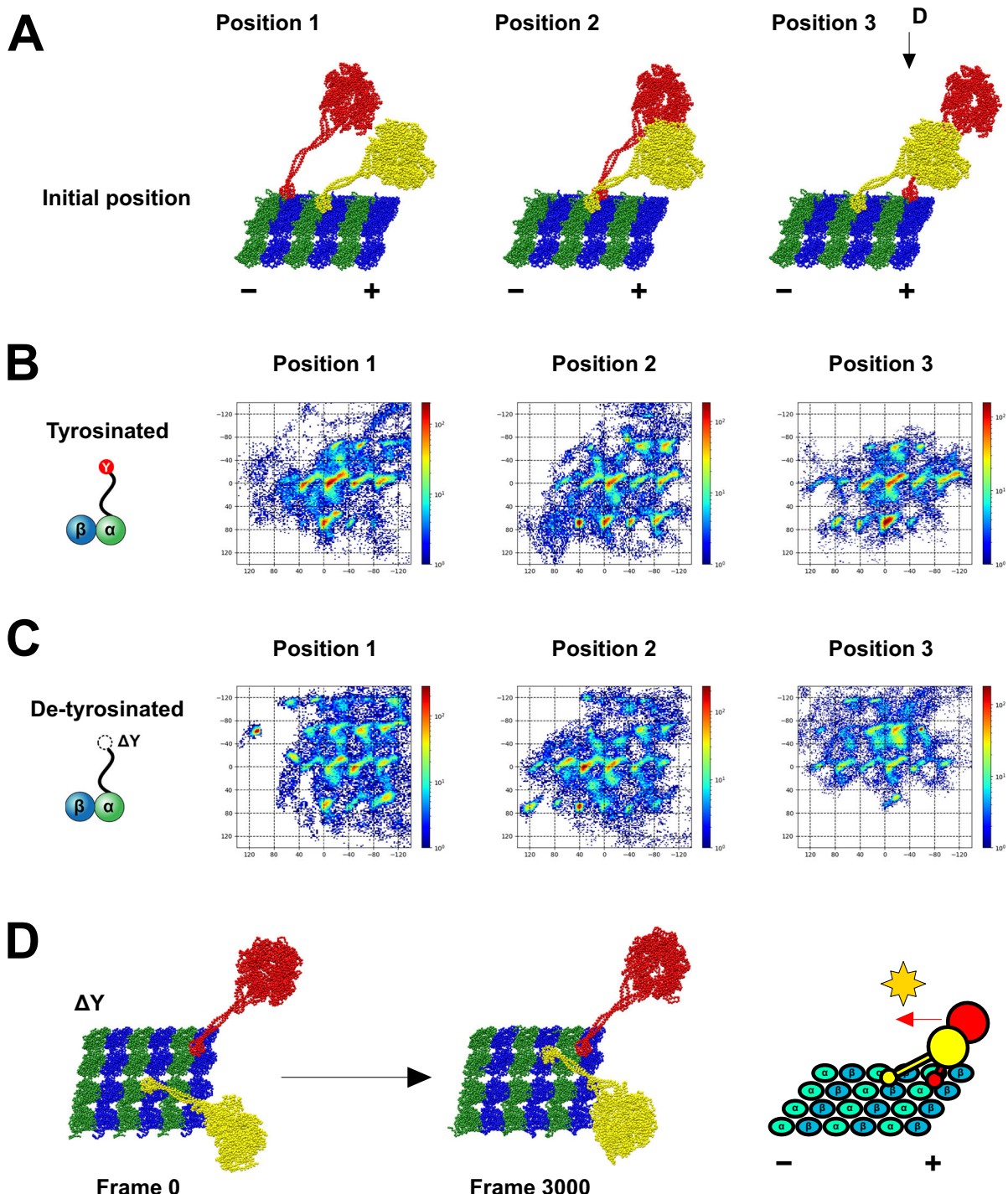

**Figure EV5. Data related to MD simulations of dynein2 dimer.**

(A) Initial structures of the MD simulations. Low-affinity dynein-2 motor domain structure (yellow) was placed at the center of the tubulin lattice with high-affinity dynein-2 (red) on the neighboring PF located at forward (position 1), adjacent (position 2), and backward (position 3). (−) and (+) indicate MT polarities. (B, C) Heatmaps of the positions of the MTBD of low-affinity dynein-2 on tubulin lattice with tyrosinated tubulins (B) and detyrosinated tubulins (C). 20 trajectories are overlaid and colored depending on the frequencies. Blue color corresponds to low frequency and red color represents higher frequency. (D) Snapshots of MD simulation results showing that the low-affinity leading head is blocking the high-affinity trailing head of dynein-2 in the detyrosinated tubulin lattice. The simulation was started from the initial state (frame 0, initial position 3 in (A)). In the simulated result (frame 3000), the low-affinity leading head moves in the direction of movement of the high-affinity trailing head. In the next cycle of the trailing head, the low-affinity leading head hinders the diffusional motion toward the minus end, and thereby, the dynein-2 dimer dissociates from the tubulin lattice (right panel). The view in (D) is indicated in (A).

