## [Peer Review File · The EMBO Journal]

Cryo-ET and MD simulations reveal that dynein-2 is tuned for binding to the A-tubule of the ciliary doublet

Haoqiang He, Shintaroh Kubo, Xuwei Chen, Qianru Lv, Azusa Kage, and Muneyoshi Ichikawa

Corresponding author(s): Muneyoshi Ichikawa (ichikawa_muneyoshi@fudan.edu.cn)

Review Timeline:

Submission Date:	6th Mar 25
Pre-decision Consultation:	25th Apr 25
Author's Revision Plan:	3rd May 25
Editorial Decision:	6th May 25
Revision Received:	8th Oct 25
Editorial Decision:	6th Nov 25
Revision Received:	10th Nov 25
Accepted:	10th Nov 25

Editor: Hartmut Vodermaier

Transaction Report:

Dear Muneyoshi,

Thank you again for submitting your manuscript for our editorial consideration. We have now received and discussed the reports of three expert referees, copied below for your information. As you will see, the reviewers appreciate the importance of the topic and acknowledge that this work may add interesting additional information complementing/extending the recent work by the Piginio group. However, they all share the same crucial concern: that tyrosination as the key factor for dynein-2 to distinguish A- from B-tubules remains to be directly demonstrated beyond mere MD simulations, in order to make this study an advance warranting publication as an EMBO Journal article. Although the referees propose some potential ways to address this, it remains unclear whether this could be decisively achieved during a limited revision period. I would therefore invite you to send me a tentative point-by-point response, detailing how you might envision addressing the key concerns of the referees in case you should be given the opportunity to revise this work for The EMBO Journal. Based on such a revision proposal, we could then discuss whether a revision for The EMBO Journal would seem realistic, or whether the study would be better candidate for rapid publication in EMBO Reports - also to ensure publication not too long after the Chhatre et al. work.

It would be great if you could get back to me with such a revision plan in the first week of May.

Looking forward to hearing from you,

Best regards,

Hartmut

Referee #1 (Report for Author)

Overview:

In their manuscript, He et al. address the question of why human dynein-2, the dynein that

drives intraflagella transport along axonemes, preferentially walks on the A-tubule, rather than the B-tubule of microtubule doublets from axonemes. They perform pelleting assays to show that the microtubule binding domain (MTBD) of dynein-2 binds more tightly to doublet microtubules from axonemes than dynein-1 does. They use EM studies to provide evidence that dynein-2 preferentially binds the A-tubule over the B-tubule. Finally, they use molecular dynamics to suggest this A-tubule specificity is linked to tyrosination.

It is not clear to me whether the data presented conclusively demonstrate that tyrosination is responsible for dynein-2's preference for A-tubules.

Major Comments:

1) The conclusion that dynein-2 preferentially binds the A-tubule is based on an EM analysis of 22 microtubules and no statistics are included in the analysis. I suggest the authors use at least three independent preparations of doublets and determine the ratio of number of dyneins on the A-tubule vs B-tubule. This will allow them to show there are significant enrichment of dynein-2 on one doublet vs the other.

2) The conclusion that tyrosination is responsible needs to be demonstrated experimentally. One possibility is to use an artificial system where tyrosination can be controlled (for example as in McKenney et al 2016). Another would be to treat the microtubule doublets with a detyrosinase (vasohibins or MATCAP) and show that this changes the distribution of dynein-2 binding.

3) Molecular Dynamics Simulations: The authors should include more details of the MD setup to make the results part clearer, such as solvent parameters (as MT-dynein interactions are electrostatically driven), number of independent simulations per conditions etc.

Minor Concerns

There are a few places in the text which could benefit from better flow - currently some of these sentences are confusing:

Lines 23-24: "suggesting a specialized regulatory mechanism involving dynein-2." - perhaps something more specific, such as 'a mechanism regulating directionality'?

Line 43: "incomplete B-tubule" could be explained.

Line 47: Why is "Nexin" capitalised?

Lines 90-91: How much tubulin is tyrosinated on the A-tubule? Does the B-tubule have no tyrosination at all? Could be specified if possible.

Lines 91-93: "Very recently, the tyrosination/detyrosination balance of the doublet was shown to be important for the proper sorting of IFT trains (Chhatre et al., 2024)" - perhaps more specifically why it is important for sorting?

Line 96: Missing an article: "an" in vitro reconstitution system.

Line's 98-99: "dynein-2 exhibits distinct binding properties to doublet microtubules compared to dynein-1" - this sentence could be taken to imply binding properties other to doublet preference, so perhaps this should be clarified.

Line 113-114: "the surface charge of the MTBD in dynein- 2 was notably distinct from that in dynein-1" - perhaps a bit more detail into how the surface charge of dynein-2 is distinct from dynein-1?

Line 117: Same as the previous point: if the fact that conserved residues are distinct between dyneins -1 and -2 is mentioned, it could be discussed more specifically.

Line 131 (subheading): "Dynein-2 is tuned" seems repetitive and too similar to the previous subheading. Perhaps it could be changed to "Dynein-2 preferentially binds A-tubules"?

Line 154: Unclear what "continued clustering" means.

Line 155: "clustering of ... were shorter and often detaching" - the sentence is grammatically incorrect.

Line 177: The term "overflow" is not explained, but it is used as a readout term for MD simulations data.

Line 189: "distinct binding properties" seems too general.

Line 206: "Dynein-2 has a distinct charge" - distinct how?

Referee #2 (Report for Author)

IFT is a motor proteins complex, which transports cilia component from cytoplasm to the ciliary tip. It is driven by kinesin toward the tip (anterograde transport) and by dynein-2 from the tip (retrograde transport). While the pioneering work by the Pigo group has revealed that anterograde and retrograde IFTs move on the B- and A-tubules of the axoneme, respectively, it was still not clear how IFTs recognize the right track to move without collision. In this work, He et al., use biochemical assays, cryo-ET and molecular dynamics, to characterize the important role of tyrosination on the doublet microtubule for recognition by dynein-2.

They first demonstrated higher affinity of the microtubule binding domain of dynein-2 to the ciliary doublet microtubules, compared to dynein-1, by binding assays. Next they engineered dimeric dynein-2 heads and employed cryo-electron tomography to visualize their binding on doublet microtubules. With this experiment they successfully proved dynein-2 binding selectively to the A-tubule. They hypothesized this biased affinity is caused by the specific tyrosination at the A-tubule. They conducted Molecular Dynamics

study on dynein-2 dimer binding to tyrosinated and detyrosinated microtubules, which supported this model.

The manuscript is compactly composed and clearly states the research and the logic. While a previous report by the Pigino group (Chhatre et al. 2025) addressed similar questions employing in vitro motility assay, this work provides complementary information by binding assays, in which they uniquely discuss that dynein-2 dissociation is the stage to differentiate binding to A- and B-tubules, cryo-ET and MD. However, the current manuscript, although successfully proving dynein-2 binding to the A-tubule with higher affinity, is still not convincing enough (except for MD simulation) about the key role of tyrosination for dynein-2 to distinguish A- and B-tubules. This reviewer would suggest possible experimental approaches to strengthen their conclusion. This reviewer believes this work will deserve publication in the EMBO Journal, once the manuscript is improved. One weakness to conclude that differential tyrosination is the key to distinguish A- and B-tubule, is the possibility that other factors, such as differences in the tubulin lattice between A- and B-tubules, are the key for high affinity of dynein-2 to the A-tubule, not (or not only) tyrosination. This can be examined by binding assay (Fig.2E; possibly quantization by scanning the gel will help) of dynein-2 to pure (reconstructed) MT, tyrosinated MT or detyrosinated doublets.

Their MD simulation is well organized, but has a gap from the experiment. It will be helpful to classify dynein dimers in cryo-ET (Fig.3FG) into the three cases shown in Fig.4A. If the population of the three positions in the tomogram matches with prediction by Fig.4CD, it will be a strong evidence that MD simulation describes the reality of dynein-2 binding.

Small points:

Typos in Line 48 "(Walton et al. 2023). " and in "cen-ter-of gravity"

Line 106 "MTBD structures": reference will be needed.

Referee #3 (Report for Author)

This is a well-structured and generally well-written manuscript addressing a significant and unresolved question in cilia biology: the molecular mechanism underlying the sorting of retrograde intraflagellar transport (IFT) driven by dynein-2 onto the A-tubule of the ciliary doublet. The authors combine biochemical assays, state-of-the-art cryo-electron tomography (cryo-ET) with Volta Phase Plate (VPP), and molecular dynamics (MD) simulations to provide evidence that dynein-2 is intrinsically tuned for binding to the A-tubule, and that this preference is mediated by the tyrosination state of α -tubulin.

While it is known that retrograde IFT occurs on the A-tubule and anterograde on the B-tubule, and that A-tubules are enriched in tyrosinated tubulin while B-tubules are enriched in detyrosinated tubulin, the precise molecular mechanism linking motor preference to

track PTMs is unclear. The paper thus advances our understanding here. However, the paper would be significantly strengthened by more direct evidence for the mechanism of tyrosine binding. Did the authors attempt 3D reconstructions from their data and fitting of molecular structures? While the MD simulations provide valuable computational evidence supporting the effect of tyrosination on binding stability, the precise structural basis for this effect (i.e., the specific interactions involving the tyrosine residue) is not revealed. The current data appears to be of good quality but without additional data or analysis that gets us closer to mechanistic insights, the paper may be better suited for EMBO Reports.

Minor corrections:

Page 3, Line 78: "polyglycyration" -> "polyglycylation".

Line 72-73: "doublet serves as dual railway tracks" - Should be "doublets serve as dual railway tracks" or "the doublet serves as a dual railway track".

Line 356: "degree" -> "degrees"

Line 357: Dose units: "79.7 electrons" - usually expressed as electrons/Å². Please clarify or confirm this is total dose over the area.

Line 376: "an MD simulations" -> "MD simulations".

Line 531: Check "CLUSTAL_X" vs "ClustalW" (Methods use W).

Figure 4: "cen-ter-of-gravity" -> "center-of-gravity".

Fudan University

Dr. Hartmut Vodermaier,
Senior Editor of *The EMBO Journal*

May 3rd, 2025

Revision Plan

We are grateful for the opportunity to revise our manuscript entitled “*Dynein-2 is tuned for the A-tubule of the ciliary doublet through tubulin tyrosination*” submitted to *The EMBO Journal* (EMBOJ-2025-120686). We would like to thank the editorial board and the reviewers for acknowledging the importance of our study and for the constructive and thoughtful feedback. As the reviewers pointed out, the major shortcoming of the current manuscript is the lack of direct experimental evidence supporting dynein-2’s preference for tyrosinated and detyrosinated microtubules. While awaiting the editorial decision, we have obtained *Chlamydomonas* cells of both wild-type (WT) and the *VashL* mutant, which lacks the tubulin detyrosination enzyme and is therefore enriched in tyrosinated tubulin within ciliary doublet microtubules (doublets). As shown in the figure below, we have confirmed that doublets can be successfully purified from both WT and *VashL* mutant cells. We are also currently obtaining the plasmid for the expression of detyrosinase to prepare doublets enriched in detyrosinated tubulin. We plan to perform binding assays, cryo-electron microscopy (cryo-EM), and cryo-electron tomography (cryo-ET) using purified dynein-2 and the doublets with tyrosination or detyrosination enriched to compare binding under matched conditions directly. In addition, we intend to conduct an additional mass spectrometry analysis, and revise the manuscript to address the reviewers’ detailed comments. A point-by-point response is attached below.

Sincerely yours,

Muneyoshi ICHIKAWA, PhD.

Tenure-track Professor

State Key Laboratory of Genetic Engineering,

Department of Biochemistry and Biophysics,

School of Life Sciences,

Fudan University, Shanghai 200438, China.

Phone: (+86)-2131246507

E-mail: ichikawa_muneyoshi@fudan.edu.cn

Figure

Figure. Preparation of the doublets from WT and *VashL* mutant *Chlamydomonas* flagella.

A. Sequential purification of the doublets from *Chlamydomonas* flagella. During the purification process, associated proteins including axonemal dyneins were depleted, resulting in clean preparations of doublets.

B. Comparison of the doublets purified from WT and *VashL* mutant *Chlamydomonas*. Doublets of similar quality were obtained, as confirmed by SDS-PAGE.

C. Representative EM images of negatively stained doublets from WT or *VashL* mutant cells. Similar qualities of the doublets (bundled) were observed in both cases. The bundled doublets will be separated by sonication and used for binding assays, cryo-EM & cryo-ET observations.

Point-by-point response

Referee #1 (Report for Author)

Overview:

In their manuscript, He et al. address the question of why human dynein-2, the dynein that drives intraflagella transport along axonemes, preferentially walks on the A-tubule, rather than the B-tubule of microtubule doublets from axonemes. They perform pelleting assays to show that the microtubule binding domain (MTBD) of dynein-2 binds more tightly to doublet microtubules from axonemes than dynein-1 does. They use EM studies to provide evidence that dynein-2 preferentially binds the A-tubule over the B-tubule. Finally, they use molecular dynamics to suggest this A-tubule specificity is linked to tyrosination.

It is not clear to me whether the data presented conclusively demonstrate that tyrosination is responsible for dynein-2's preference for A-tubules.

Major Comments:

1) The conclusion that dynein-2 preferentially binds the A-tubule is based on an EM analysis of 22 microtubules and no statistics are included in the analysis. I suggest the authors use at least three independent preparations of doublets and determine the ratio of number of dyneins on the A-tubule vs B-tubule. This will allow them to show there are significant enrichment of dynein-2 on one doublet vs the other.

We will prepare doublets from three independent preparations, incubate with dynein-2, and perform cryo-EM and cryo-ET analyses to confirm whether there are significant differences between A- and B-tubule binding.

2) The conclusion that tyrosination is responsible needs to be demonstrated experimentally. One possibility is to use an artificial system where tyrosination can be controlled (for example as in McKenney et al 2016). Another would be to treat the microtubule doublets with a detyrosinase (vasohibins or MATCAP) and show that this changes the distribution of dynein-2 binding.

To perform experiments using samples with more controlled levels of tyrosination and detyrosination, we plan to use three types of doublets: (1) detyrosinase-treated doublets (enriched in detyrosinated tubulin), (2) untreated WT doublets (containing both tyrosinated and detyrosinated tubulin), and (3) doublets from *VashL* mutant strains (enriched in tyrosinated tubulin). By comparing these results, we aim to provide more conclusive evidence for the role of tyrosination.

3) Molecular Dynamics Simulations: The authors should include more details of the MD setup to make the results part clearer, such as solvent parameters (as MT-dynein interactions are electrostatically driven), number of independent simulations per conditions etc.

We appreciate the reviewer for this important comment. In our MD simulations, we used an implicit solvent model with a relative dielectric constant of 78, ionic strength of 0.1 M, and temperature set at 300 K. The equations of motion were integrated using underdamped Langevin dynamics. For each condition, we performed 20 independent simulations with different random seeds. We prepared three distinct initial dynein configurations and thus conducted a total of 60 simulations for both the tyrosinated and detyrosinated microtubules. These methodological details will be added to the revised manuscript.

Minor Concerns

There are a few places in the text which could benefit from better flow - currently some of these sentences are confusing:

Lines 23-24: "suggesting a specialized regulatory mechanism involving dynein-2." - perhaps something more specific, such as 'a mechanism regulating directionality'?

We appreciate that the reviewer pointed out the confusing parts. We will revise the expression in the revised manuscript.

Line 43: "incomplete B-tubule" could be explained.

We will include more details in the revised manuscript.

Line 47: Why is "Nexin" capitalised?

We will fix this in the revised manuscript.

Lines 90-91: How much tubulin is tyrosinated on the A-tubule? Does the B-tubule have no tyrosination at all? Could be specified if possible.

We intend to perform mass spectrometry analysis to quantify and compare the tyrosinated and detyrosinated tubulin levels in the samples.

Lines 91-93: "Very recently, the tyrosination/detyrosination balance of the doublet was shown to be important for the proper sorting of IFT trains (Chhatre et al., 2024)" - perhaps more specifically why it is important for sorting?

We will modify the sentence to include more details for this point.

Line 96: Missing an article: "an" in vitro reconstitution system.

We will fix this point in the revised manuscript.

Line's 98-99: "dynein-2 exhibits distinct binding properties to doublet microtubules compared to dynein-1" - this sentence could be taken to imply binding properties other to doublet preference, so perhaps this should be clarified.

We will revise the sentence to describe the binding properties related to doublet preference more accurately.

Line 113-114: "the surface charge of the MTBD in dynein- 2 was notably distinct from that in dynein-1" - perhaps a bit more detail into how the surface charge of dynein-2 is distinct from dynein-1?

We will include more details about this point in the revised manuscript.

Line 117: Same as the previous point: if the fact that conserved residues are distinct between dyneins -1 and -2 is mentioned, it could be discussed more specifically.

We will discuss this point more specifically in the revised manuscript.

Line 131 (subheading): "Dynein-2 is tuned" seems repetitive and too similar to the previous subheading. Perhaps it could be changed to "Dynein-2 preferentially binds A-tubules"?

We will change the subheading to avoid similarity to the previous subheading.

Line 154: Unclear what "continued clustering" means.

We will define the term or fix the sentence to make it clearer.

Line 155: "clustering of... were shorter and often detaching" - the sentence is grammatically incorrect.

We will correct the grammar of this sentence in the revised manuscript.

Line 177: The term "overflow" is not explained, but it is used as a readout term for MD simulations data.

We will clearly explain this point in the revised manuscript.

Line 189: "distinct binding properties" seems too general.

We will discuss this in more detail in the revised manuscript.

Line 206: "Dynein-2 has a distinct charge" - distinct how?

We will describe in more detail regarding this point in the revised manuscript.

Referee #2 (Report for Author)

IFT is a motor proteins complex, which transports cilia component from cytoplasm to the ciliary tip. It is driven by kinesin toward the tip (anterograde transport) and by dynein-2 from the tip (retrograde transport). While the pioneering work by the Pigino group has revealed that anterograde and retrograde IFTs move on the B- and A-tubules of the axoneme, respectively, it was still not clear how IFTs recognize the right track to move without collision. In this work, He et al., use biochemical assays, cryo-ET and molecular dynamics, to characterize the important role of tyrosination on the doublet microtubule for recognition by dynein-2.

They first demonstrated higher affinity of the microtubule binding domain of dynein-2 to the ciliary doublet microtubules, compared to dynein-1, by binding assays. Next they engineered dimeric dynein-2 heads and employed cryo-electron tomography to visualize their binding on doublet microtubules. With this experiment they successfully proved dynein-2 binding selectively to the A-tubule. They hypothesized this biased affinity is caused by the specific tyrosination at the A-tubule. They conducted Molecular Dynamics study on dynein-2 dimer binding to tyrosinated and detyrosinated microtubules, which supported this model.

The manuscript is compactly composed and clearly states the research and the logic. While a previous report by the Pigino group (Chhatre et al. 2025) addressed similar questions employing in vitro motility assay, this work provides complementary information by binding assays, in which they uniquely discuss that dynein-2 dissociation is the stage to differentiate binding to A- and B-tubules, cryo-ET and MD.

We appreciate the reviewer's thorough and encouraging assessment of our study. We are especially grateful for the recognition that our work complements the recent findings by the Pigino group and provides a new perspective through biochemical assays and cryo-ET.

However, the current manuscript, although successfully proving dynein-2 binding to the A-tubule with higher affinity, is still not convincing enough (except for MD simulation) about the key role of tyrosination for dynein-2 to distinguish A- and B-tubules. This reviewer would suggest possible experimental approaches to strengthen their conclusion. This reviewer believes this work will deserve publication in the EMBO Journal, once the manuscript is improved.

One weakness to conclude that differential tyrosination is the key to distinguish A- and B-tubule, is the possibility that other factors, such as differences in the tubulin lattice between A- and B-tubules, are the key for high affinity of dynein-2 to the A-tubule, not (or not only) tyrosination. This can be examined by binding assay (Fig.2E; possibly quantization by scanning the gel will help) of dynein-2 to pure (reconstructed) MT, tyrosinated MT or detyrosinated doublets.

Thank you for pointing this out. We agree that this is a critical issue to address. To assess the contribution of tubulin tyrosination more directly, we plan to compare dynein-2 binding to (1) doublets treated with recombinant detyrosinase (enriched in detyrosinated tubulin), (2) untreated WT doublets (containing both tyrosinated and detyrosinated tubulin), and (3) doublets from mutant strain lacking detyrosinase (enriched in tyrosinated tubulin).

Their MD simulation is well organized, but has a gap from the experiment. It will be helpful to classify dynein dimers in cryo-ET (Fig.3FG) into the three cases shown in Fig.4A. If the population of the three positions in the tomogram matches with prediction by Fig.4CD, it will be a strong evidence that MD simulation describes the reality of dynein-2 binding.

Thank you for commending our MD simulation results. We will analyze the cryo-ET results and see if we can correlate the dynein-2 dimer composition with MD simulation results.

Small points:

Typos in Line 48 "(Walton et al. 2023). " and in "cen-ter-of gravity"

We will fix these typos in the revised manuscript.

Line 106 "MTBD structures": reference will be needed.

We will add references for the sentences.

Referee #3 (Report for Author)

This is a well-structured and generally well-written manuscript addressing a significant and unresolved question in cilia biology: the molecular mechanism underlying the sorting of retrograde intraflagellar transport (IFT) driven by dynein-2 onto the A-tubule of the ciliary doublet. The authors combine biochemical assays, state-of-the-art cryo-electron tomography (cryo-ET) with Volta Phase Plate (VPP), and molecular dynamics (MD) simulations to provide evidence that dynein-2 is intrinsically tuned for binding to the A-tubule, and that this preference is mediated by the tyrosination state of α -tubulin.

We sincerely appreciate this reviewer's positive and encouraging comments. We are glad that the manuscript was found to be well-structured and that the importance of the biological question and the integrated approach combining biochemistry, cryo-ET, and MD simulations was recognized.

While it is known that retrograde IFT occurs on the A-tubule and anterograde on the B-tubule, and that A-tubules are enriched in tyrosinated tubulin while B-tubules are enriched in detyrosinated tubulin, the precise molecular mechanism linking motor preference to track PTMs is unclear. The paper thus advances our understanding here.

We are grateful that the reviewer acknowledged the contribution of our work toward advancing the understanding of how motor preference is linked to tubulin post-translational modifications.

However, the paper would be significantly strengthened by more direct evidence for the mechanism of tyrosine binding. Did the authors attempt 3D reconstructions from their data and fitting of molecular structures? While the MD simulations provide valuable computational evidence supporting the effect of tyrosination on binding stability, the precise structural basis for this effect (i.e., the specific interactions involving the tyrosine residue) is not revealed. The current data appears to be of good quality but without additional data or analysis that gets us closer to mechanistic insights, the paper may be better suited for EMBO Reports.

We appreciate the reviewer's insightful comment. While the current resolution does not allow unambiguous identification of side-chain level interactions, we will reanalyze the cryo-ET data and assess whether available structural models of dynein-2 can be fitted into the density maps. In addition, to obtain more direct evidence for the mechanism, we plan to perform biochemical, cryo-EM, and cryo-ET analyses using doublets with more controlled levels of tyrosination and detyrosination. Furthermore, we will analyze the residue-level contact frequencies between dynein and either α - or β -tubulin during the MD simulations. By comparing the tyrosinated and detyrosinated conditions in the MD simulations, we aim to identify specific contact patterns that underlie the distinct binding behaviors, and we will report these findings in the revised manuscript.

Minor corrections:

Page 3, Line 78: "polyglycyration" -> "polyglycylation".

We apologize for the typo. We will fix it in the revised manuscript.

Line 72-73: "doublet serves as dual railway tracks" - Should be "doublets serve as dual railway tracks" or "the doublet serves as a dual railway track".

The sentence will be fixed in the revised manuscript.

Line 356: "degree" -> "degrees"

We will fix it in the revised manuscript.

Line 357: Dose units: "79.7 electrons" - usually expressed as electrons/Å². Please clarify or confirm this is total dose over the area.

We apologize for the mistake. We will fix it to electrons/ Å² in the revised manuscript.

Line 376: "an MD simulations" -> "MD simulations".

We will fix it in the revised manuscript.

Line 531: Check "CLUSTAL_X" vs "ClustalW" (Methods use W).

We will fix this point in the revised manuscript.

Figure 4: "cen-ter-of-gravity" -> "center-of-gravity".

We will fix this typo in the revised manuscript.

Dr. Muneyoshi Ichikawa
Fudan University
2005 hao, Songhu-lu, Yangpu-qu
Shanghai, Shanghai 200438
China

6th May 2025

Re: EMBOJ-2025-120686
Dynein-2 is tuned for the A-tubule of the ciliary doublet through tubulin tyrosination

Dear Muneyoshi,

Thank you for sending me your revision plan for your study on dynein-2 tracking via tyrosinated A-tubules. I have now had a chance to go through them, and concluded that a revised manuscript may well become suitable for EMBO Journal publication, provided that the planned revision experiments (especially for comparing three types of doublets differing in their tyrosination) should turn out successful. Likewise, also the other key revisions regarding further structural and biochemical analyses and improved MD descriptions should be very helpful. I am therefore happy to formally invite you to prepare and resubmit a new version of the study, revised along the lines proposed in your letter.

Please keep in mind that it is our policy to allow only a single round of (major) revision, and please do update me should there be any unexpected problems with the revisions, or should you require an extension beyond the default 3-months deadline. As always, competing manuscript published during the course of this revision will not affect our final decision on your study. Finally, please note the detailed information and guidelines on how to prepare a revision below (and in our online Guide to Authors) - closely adhering to them shall greatly facilitate the editorial process at the time of resubmission.

Thank you again for the opportunity to consider this work, and I look forward to receiving your revision in due time.

With kind regards,

Hartmut

9) To facilitate reproducibility and cross-laboratory adoption of methodologies, please structure the Materials & Methods section as outlined in our guide to authors, including a completed Reagents and Tools Table that can be downloaded from our author guidelines as well (<https://www.embopress.org/page/journal/14602075/authorguide#structuredmethods>).

10) Digital image enhancement is acceptable practice, as long as it accurately represents the original data and conforms to community standards. If a figure has been subjected to significant electronic manipulation, this must be clearly noted in the figure legend and/or the 'Materials and Methods' section. The editors reserve the right to request original versions of figures and the original images that were used to assemble the figure. Finally, we generally encourage uploading of numerical as well as gel/blot image source data; for details see: embopress.org/page/journal/14602075/authorguide#sourcedata

In the interest of ensuring the conceptual advance provided by the work, we recommend submitting a revision within 3 months (4th Aug 2025). Please discuss the revision progress ahead of this time with the editor if you require more time to complete the revisions. Use the link below to submit your revision:

Link Not Available

Point-by-point response

Referee #1 (Report for Author)

Overview:

In their manuscript, He et al. address the question of why human dynein-2, the dynein that drives intraflagella transport along axonemes, preferentially walks on the A-tubule, rather than the B-tubule of microtubule doublets from axonemes. They perform pelleting assays to show that the microtubule binding domain (MTBD) of dynein-2 binds more tightly to doublet microtubules from axonemes than dynein-1 does. They use EM studies to provide evidence that dynein-2 preferentially binds the A-tubule over the B-tubule. Finally, they use molecular dynamics to suggest this A-tubule specificity is linked to tyrosination.

It is not clear to me whether the data presented conclusively demonstrate that tyrosination is responsible for dynein-2's preference for A-tubules.

Major Comments:

1) The conclusion that dynein-2 preferentially binds the A-tubule is based on an EM analysis of 22 microtubules and no statistics are included in the analysis. I suggest the authors use at least three independent preparations of doublets and determine the ratio of number of dyneins on the A-tubule vs B-tubule. This will allow them to show there are significant enrichment of dynein-2 on one doublet vs the other.

We greatly appreciate the reviewer's insightful comment on this point. Although we were not able to perform three independent cryo-ET observations due to the limited time for revision, we have substantially revised our manuscript as below.

First, to show reproducibility, we included a new dataset (dataset 2) obtained by cryo-ET of an *in vitro* reconstituted sample in the revised manuscript, showing similar A-tubule accumulation (Fig EV4B). In addition, we have confirmed that the binding of GST-Dyn2 molecules to one side of the doublet, presumably the A-tubule, is reproducible in *Chlamydomonas* doublets, as shown in Fig EV4C. These points are included in the main text (Lines: 172-174).

Second, to address the issue of a lack of statistical analysis, we have quantified the number of dynein-2 heads bound to either the A-tubule or the B-tubule side, calculated the number of heads per 100 nm, and performed statistical analyses (Fig 3H). Both dataset 1 (original dataset) and dataset 2 showed

significantly more binding to the A-tubule sides. These points were included in the main text (Lines 168-170).

Finally, the manuscript was revised so that the intensions are more clearly conveyed. Since some of the protofilaments (PFs) in the A-tubule are not exposed and therefore inaccessible to dynein-2, whereas all the PFs in the B-tubule are accessible, the actual difference in binding is likely greater than what is observed. This point was also included in the revised manuscript (Lines 170-172). We failed to mention this in the previous version of the manuscript, but all doublets shown in Fig 3E have more binding on the A-tubule sides. The panel in Fig 3E was replaced with a higher-quality one so that the readers can acknowledge this point. This point was also added in the figure legend of Fig 3E.

We hope these changes sufficiently support our conclusion.

2) The conclusion that tyrosination is responsible needs to be demonstrated experimentally. One possibility is to use an artificial system where tyrosination can be controlled (for example as in McKenney et al 2016). Another would be to treat the microtubule doublets with a detyrosinase (vasohibins or MATCAP) and show that this changes the distribution of dynein-2 binding.

We appreciate the reviewer's critical suggestion. To address this point, we have performed decoration experiments using doublet microtubules purified from *VashL Chlamydomonas* mutant, which lacks detyrosinase. In the *VashL* mutant doublet, A- and B-tubules are expected to be tyrosinated similarly. So far, we have not observed the accumulation of GST-Dyn2 molecules on one side of the *VashL* doublets, unlike WT doublets. However, we feel that this point deserves more thorough analysis and validation before publication, and we would like to pursue this point further in future research.

In the revised manuscript, we have clearly distinguished between experimentally obtained findings and those based on MD simulations and toned down our conclusions. We have also modified the title to "Cryo-ET and MD simulations reveal that dynein-2 is tuned for binding to the A-tubule of the ciliary doublet" so that it would not be an overstatement.

Furthermore, to support our conclusion, we have also performed new contact analysis in MD simulations. Although this is an in silico analysis, we observed distinct interacting patterns of dynein-2 MTBD and tubulin dimer depending on the presence or absence of the C-terminal tyrosine residue of α -tubulin (Fig 5E and F). The sentences regarding this analysis were included in the main text (Lines 243-255 and Lines 301-311).

We hope these changes sufficiently address the reviewer's concern.

3) Molecular Dynamics Simulations: The authors should include more details of the MD setup to make the results part clearer; such as solvent parameters (as MT-dynein interactions are electrostatically driven), number of independent simulations per conditions etc.

We appreciate the reviewer for this important comment. In our MD simulations, we used an implicit solvent model with a relative dielectric constant of 78, ionic strength of 0.1 M, and a temperature of 300 K. The equations of motion were integrated using underdamped Langevin dynamics. For each condition, we performed 20 independent simulations with different random seeds. We prepared three distinct initial dynein configurations, and thus conducted a total of 60 simulations for both the tyrosinated and detyrosinated microtubules. To reduce dependence on the initial configuration, the first 5×10^6 MD steps were excluded from the analysis. These methodological details have been added to the Materials and Methods section in the revised manuscript (Lines 636-645 and Lines 650-651). In the previous version of the manuscript, the temperature was incorrectly written as 323 K, and it has now been corrected. We apologize for the mistake.

Minor Concerns

There are a few places in the text which could benefit from better flow - currently some of these sentences are confusing:

We appreciate that the reviewer pointed out the confusing parts. We have fixed these points as below.

Lines 23-24: "suggesting a specialized regulatory mechanism involving dynein-2." - perhaps something more specific, such as 'a mechanism regulating directionality'?

We have fixed this sentence to make it more specific (Lines 24-25). Due to the word limitation of the Abstract section, other parts of the Abstract have also been modified to accommodate the change. To better show what was not clear in the previous reports, we have also updated Fig 1.

Line 43: "incomplete B-tubule" could be explained.

We have modified the sentences to explain more details about the organization of doublets (Lines 41-45). We have also updated Fig 1 and mentioned it in these sentences so that the readers can refer to the figure.

Line 47: Why is "Nexin" capitalised?

We have fixed this point in the revised manuscript (Line 48).

Lines 90-91: How much tubulin is tyrosinated on the A-tubule? Does the B-tubule have no tyrosination at all? Could be specified if possible.

We appreciate this insightful comment. We have performed totally new mass spectrometry analyses of microtubule fraction purified from *Chlamydomonas* flagella for this revision, quantifying the proportions of tyrosinated and detyrosinated α -tubulin C-terminal peptides.

We found that tyrosinated α -tubulin C-terminal peptides accounted for only 33.7% of the total peptides in the WT sample, despite the A-tubule having more protofilaments compared with the B-tubule. We used the *VashL* mutant, which lacks the tubulin detyrosinase, as a control, and detected only tyrosinated peptides in the *VashL* doublets. Our results suggest that only a portion of the A-tubule is tyrosinated, and the B-tubule is mostly detyrosinated.

Furthermore, our mass spectrometry results showed that glutamylated α -tubulin accounted for 6.35% of total tubulin peptides. This is consistent with a recent report showing that only one PF is glutamylated (Alvarez Viar et al., 2024), confirming our strategy.

The tyrosinated fraction of PFs likely localizes at the pathway of the retrograde IFT. We have included these results as **Figure EV3**, and included the paragraphs regarding the MS analysis results in the Results section (**Lines 190-211**) and Discussion section (**Lines 277-287**). We believe these new results provide important new insights into the IFT regulatory mechanism as well as the spatial patterning of post-translational modifications in ciliary doublets.

Lines 91-93: "Very recently, the tyrosination/detyrosination balance of the doublet was shown to be important for the proper sorting of IFT trains (Chhatre et al., 2024)" - perhaps more specifically why it is important for sorting?

We have added more details for this point in the revised manuscript (**Lines 94-98**).

Line 96: Missing an article: "an" in vitro reconstitution system.

We have fixed this point in the revised manuscript (**Line 102**).

Line's 98-99: "dynein-2 exhibits distinct binding properties to doublet microtubules compared to dynein-1" - this sentence could be taken to imply binding properties other to doublet preference, so perhaps this should be clarified.

We have modified this part so that the intention would be clearer for the reader (**Lines 100-101**). The sentences were also modified so that it would be clear which parts of the conclusions are based on experimental evidence.

Line 113-114: "the surface charge of the MTBD in dynein- 2 was notably distinct from that in dynein-1" - perhaps a bit more detail into how the surface charge of dynein-2 is distinct from dynein-1?

We included more details about the surface charge in the revised manuscript (Lines 119-122).

Line 117: Same as the previous point: if the fact that conserved residues are distinct between dyneins -1 and -2 is mentioned, it could be discussed more specifically.

We have added sentences regarding this point (Lines 125-131). Furthermore, we have added white arrowheads to Fig 2C and Fig EV1 to clearly highlight the residues in the figure and added sentences in the manuscript accordingly (Line 125) as well as in the figure legends.

Line 131 (subheading): "Dynein-2 is tuned" seems repetitive and too similar to the previous subheading. Perhaps it could be changed to "Dynein-2 preferentially binds A-tubules"?

We appreciate for your comment. We have fixed this point in the revised manuscript (Line 145).

Line 154: Unclear what "continued clustering" means.

Line 155: "clustering of... were shorter and often detaching" - the sentence is grammatically incorrect.

We apologize for the confusing expression. We meant the row of the GST-Dyn2 molecules by clustering. Since we included quantification results of head numbers of GST-Dyn2 per 100 nm in the revised manuscript, we felt that the part regarding clustering is not as objective as the quantification result. Therefore, we have removed the part of clustering from the revised manuscript. For Fig 3G, we have replaced the panel with doublets with GST-Dyn2 molecules, where each molecule can be distinguished, since this is the condition under which quantification was performed in Fig 3H.

Line 177: The term "overflow" is not explained, but it is used as a readout term for MD simulations data.

We apologize for the lack of explanation. We have included the definition of overflow in the revised manuscript (Lines 228-231).

Line 189: "distinct binding properties" seems too general.

Thank you for pointing this out. We revised the sentence to make it more specific (Lines 258-259).

Line 206: "Dynein-2 has a distinct charge" - distinct how?

We apologize for the lack of details. We have described this point in more detail in the revised manuscript (Lines 292-294).

Referee #2 (Report for Author)

IFT is a motor proteins complex, which transports cilia component from cytoplasm to the ciliary tip. It is driven by kinesin toward the tip (anterograde transport) and by dynein-2 from the tip (retrograde transport). While the pioneering work by the Pigino group has revealed that anterograde and retrograde IFTs move on the B- and A-tubules of the axoneme, respectively, it was still not clear how IFTs recognize the right track to move without collision. In this work, He et al., use biochemical assays, cryo-ET and molecular dynamics, to characterize the important role of tyrosination on the doublet microtubule for recognition by dynein-2.

They first demonstrated higher affinity of the microtubule binding domain of dynein-2 to the ciliary doublet microtubules, compared to dynein-1, by binding assays. Next they engineered dimeric dynein-2 heads and employed cryo-electron tomography to visualize their binding on doublet microtubules. With this experiment they successfully proved dynein-2 binding selectively to the A-tubule. They hypothesized this biased affinity is caused by the specific tyrosination at the A-tubule. They conducted Molecular Dynamics study on dynein-2 dimer binding to tyrosinated and detyrosinated microtubules, which supported this model.

The manuscript is compactly composed and clearly states the research and the logic. While a previous report by the Pigino group (Chhatre et al. 2025) addressed similar questions employing in vitro motility assay, this work provides complementary information by binding assays, in which they uniquely discuss that dynein-2 dissociation is the stage to differentiate binding to A- and B-tubules, cryo-ET and MD.

We appreciate the reviewer's thorough and encouraging assessment of our study. We are especially grateful for the recognition that our work complements the recent findings by the Pigino group and provides a new perspective through biochemical assays and cryo-ET.

However, the current manuscript, although successfully proving dynein-2 binding to the A-tubule with higher affinity, is still not convincing enough (except for MD simulation) about the key role of tyrosination for dynein-2 to distinguish A- and B-tubules. This reviewer would suggest possible experimental approaches to strengthen their conclusion. This reviewer believes this work will deserve publication in the EMBO Journal, once the manuscript is improved.

One weakness to conclude that differential tyrosination is the key to distinguish A- and B-tubule, is the possibility that other factors, such as differences in the tubulin lattice between A- and B-tubules, are

the key for high affinity of dynein-2 to the A-tubule, not (or not only) tyrosination. This can be examined by binding assay (Fig.2E; possibly quantization by scanning the gel will help) of dynein-2 to pure (reconstructed) MT, tyrosinated MT or detyrosinated doublets.

We appreciate the reviewer's insightful comment. We attempted to reveal this point by using doublets purified from WT and *VashL* mutant *Chlamydomonas* cells. We have observed the accumulation of GST-Dyn2 molecules on one side of the *Chlamydomonas* WT doublet (Fig EV4C). In contrast, we have not observed accumulation of GST-Dyn2 molecules on one side of the *VashL* mutant doublets, which lacks detyrosinated tubulins. Although this supports that the tyrosination do affect the recruitment of GST-Dyn2 molecules, we feel that this result needs to be confirmed more carefully before publication. In the revised manuscript, therefore, we have modified the text to make it clear which parts are based on experimental data and which are based on in silico analysis. We have toned down our conclusions of the manuscript, and the title has also been revised so that it reflects the situation better.

We have also included sentences regarding the possibilities that the local curvature of the doublets could also affect the dynein-2 binding (Lines 337-350). It would require thorough analyses combining *in vitro* reconstitution and in silico analysis to assess the effects of the local deformation of the tubulin lattices for the recruitment of dynein-2 molecules. Therefore, for this point, we would like to pursue further in the future study.

We hope that these changes sufficiently answer the reviewer's concerns.

Their MD simulation is well organized, but has a gap from the experiment. It will be helpful to classify dynein dimers in cryo-ET (Fig.3FG) into the three cases shown in Fig.4A. If the population of the three positions in the tomogram matches with prediction by Fig.4CD, it will be a strong evidence that MD simulation describes the reality of dynein-2 binding.

Thank you very much for commending our MD simulation results. And thank you for pointing out the gap between our experiments and MD simulations, it was the part missing in our previous manuscript. In the revised manuscript, we tried to fill the gap between our experiment part and simulation part. We classified the dynein-2 dimer configurations from cryo-ET results and prepared a totally new figure (current Fig 4A and B). We also performed cryo-ET analysis of intact *Chlamydomonas* flagella and showed that similar configuration dynein-2 molecule is present *in situ* (current Fig 4C), supporting that the configurations we observed *in vitro* are physiological. We added sentences about these points in the Results section (Lines 175-188 and Lines 217-220).

Based on our classification of the configurations, the molecules corresponding to the Position 3, which

showed the most notable difference in the MD simulations between tyrosination and detyrosination conditions, were the majority. This point was also included in the Discussion section (Lines 313-315).

Small points:

Typos in Line 48 "(Walton et al. 2023). " and in "cen-ter-of gravity"

We apologize for the mistakes. We have fixed these points in the revised manuscript (Line 51 and Figure 5C, D legend).

Line 106 "MTBD structures": reference will be needed.

We have added PDB IDs and references in the revised manuscript (Lines 110-111).

Referee #3 (Report for Author)

This is a well-structured and generally well-written manuscript addressing a significant and unresolved question in cilia biology: the molecular mechanism underlying the sorting of retrograde intraflagellar transport (IFT) driven by dynein-2 onto the A-tubule of the ciliary doublet. The authors combine biochemical assays, state-of-the-art cryo-electron tomography (cryo-ET) with Volta Phase Plate (VPP), and molecular dynamics (MD) simulations to provide evidence that dynein-2 is intrinsically tuned for binding to the A-tubule, and that this preference is mediated by the tyrosination state of α -tubulin.

We sincerely appreciate this reviewer's positive and encouraging comments. We are pleased that the manuscript was found to be well-structured and that the importance of the biological question and the integrated approach combining biochemistry, cryo-ET, and MD simulations was recognized.

While it is known that retrograde IFT occurs on the A-tubule and anterograde on the B-tubule, and that A-tubules are enriched in tyrosinated tubulin while B-tubules are enriched in detyrosinated tubulin, the precise molecular mechanism linking motor preference to track PTMs is unclear. The paper thus advances our understanding here.

We are grateful that the reviewer acknowledged the contribution of our work toward advancing the understanding of how motor preference is linked to tubulin post-translational modifications.

However, the paper would be significantly strengthened by more direct evidence for the mechanism of tyrosine binding. Did the authors attempt 3D reconstructions from their data and fitting of molecular structures? While the MD simulations provide valuable computational evidence supporting the effect

of tyrosination on binding stability, the precise structural basis for this effect (i.e., the specific interactions involving the tyrosine residue) is not revealed. The current data appears to be of good quality but without additional data or analysis that gets us closer to mechanistic insights, the paper may be better suited for EMBO Reports.

We appreciate the reviewer's insightful comment. In the revised manuscript, we included a panel of comparison of our cryo-ET result with available dynein-2 motor domain structure as our interpretation in Fig 3G. With this result, we were able to show that the GST-Dyn2 molecules are bound to the doublet in a canonical way using their MTBDs. These points were included in the main text (Lines 166-168).

Regarding the 3D reconstruction, all the previous structural information of dynein MTBD-microtubule interface was obtained by single-particle cryo-EM (Redwine, W. B. *et al.*, *Science*, 2012; Uchimura, S. *et al.*, *Journal of Cell Biology*, 2015; Lacey, S. E. *et al.*, *eLife*, 2019; Nishida, N. *et al.*, *Nature Communications*, 2020; Rao, Q. *et al.*, *Nature Structural & Molecular Biology*, 2021). For this reason, our current cryo-ET data is not suitable to obtain high-resolution information, and obtaining a high-resolution structure of dynein-2 MTBD bound to microtubule would require further extensive structural analysis using single-particle cryo-EM. Therefore, we would like to pursue this point in future study. These points are also included in the revised manuscript (Lines 311-312).

Although it is unrealistic to obtain high-resolution structures within the limited time for revision, we performed a totally new contact map analysis in our MD simulations to identify the cause of the different behavior of dynein-2 dimer on tyrosinated and detyrosinated tubulin lattices (Fig 5E and F). This analysis allowed us to identify the interacting residues between dynein-2 MTBD and tubulin. By comparing the tyrosinated and detyrosinated tubulin results, we found that increases and decreases of contacts in the presence and absence of the tyrosine residue of the tubulin. These points have been included in the Results section (Lines 243-255) and Discussion section (Lines 301-311).

We hope that these changes are sufficient to answer to this comment.

Minor corrections:

Page 3, Line 78: "polyglycyration" -> "polyglycylation".

We apologize for the typo. We have fixed this in the revised manuscript (Line 82).

Line 72-73: "doublet serves as dual railway tracks" - Should be "doublets serve as dual railway tracks" or "the doublet serves as a dual railway track".

The sentence was fixed to "the doublet serves as a dual railway track" (Lines 76-77).

Line 356: "degree" -> "degrees"

We have fixed this in the revised manuscript (Line 498).

Line 357: Dose units: "79.7 electrons" - usually expressed as electrons/Å². Please clarify or confirm this is total dose over the area.

We apologize for the mistake. We fixed it to electrons/Å² in the revised manuscript (Line 499).

Line 376: "an MD simulations" -> "MD simulations".

We have fixed this point in the revised manuscript (Line 621).

Line 531: Check "CLUSTAL_X" vs "ClustalW" (Methods use W).

We apologize for the mistake in the reference. We have fixed the reference to correct one Thompson *et al*, (1994).

Figure 4: "cen-ter-of-gravity" -> "center-of-gravity".

We have fixed this typo in the revised manuscript (Figure 5C, D legend).

Apart from the reviewers' comments, the following points were modified in the revised manuscript.

-Mr. Xuwei Chen was added as a co-first author (third position) for his contribution to MS analysis and cryo-ET analysis of the intact flagella part for the revision.

-We have included sentences that the MTBDs of the autoinhibited form of dynein-2 are not associated with doublets while transported by anterograde IFT (Lines 73-75), since we believe this point is also important for our conclusion as explained in the Discussion section (Lines 271-272).

-The sentences regarding MD simulations were rephrased to avoid possible misunderstandings (Lines 238-239 and Line 242).

-The representative reconstituted tomogram has been deposited to Electron Microscopy Data Bank, and the sentence is added in the Data availability section.

-Stepanek & Pigino, (2016) utilized resin-embedded tomography, instead of cryo-ET. This point was

wrongly written in the previous version of the manuscript. This point has been fixed in the revised version (Lines 77 and 262-263).

-The gel image of Fig 2E was replaced with the better-quality one.

-The images for Fig 3E and G were replaced with the better-quality ones.

-Panel for MIPs, which was in the original Fig 3, was moved to Fig EV4A since there were more panels in revised Fig 3, and to improve the flow between the panels.

-The name of the affiliation 1 has been changed recently, so the affiliation part is modified.

-Minor points like figure numbering, typos, mistakes in grammar, consistency of expressions, and small mistakes in the figures were fixed as highlighted in the main text.

We want to thank the reviewers again for their time and effort to improve our manuscript.

Dr. Muneyoshi Ichikawa
Fudan University
Department of Biochemistry and Biophysics
2005 hao, Songhu-lu, Yangpu-qu
Shanghai, Shanghai 200438
China

6th Nov 2025

Re: EMBOJ-2025-120686R

Cryo-ET and MD simulations reveal that dynein-2 is tuned for binding to the A-tubule of the ciliary doublet

Dear Muneyoshi,

Thank you for submitting your revised manuscript to The EMBO Journal. It has now been re-reviewed by original referees 2 and 3, who both acknowledged the improvements to the study. While referee 2 would ideally still want more direct evidence for tyrosination-mediated dynein-2 targeting, I agree with referee 3 that this concern is outweighed by the various other pieces of evidence added during the revision, and would therefore not see it warranted to unnecessarily delay publication of this work any further.

Prior to formal acceptance, there are still some editorial issues that would need to be urgently addressed:

- Please upload all main Figures and all Expanded View figures as image rather than PPTX files, with sufficient resolution/quality for production.

- Please adjust the order of the manuscript sections, and also make sure to use the correct section headers: Title page with complete author information, Abstract, Keywords, Introduction, Results, Discussion, Methods, Data Availability, Acknowledgements, Disclosure and Competing Interests Statement, References, Main Figure Legends, Tables, Expanded Figure Legends.

Please remove the duplicated "competing interest" statement from the title page.

- On the abstract page of the manuscript, please include 4-5 general keyword terms to enhance searchability.

- Please carefully go through the reference list and make sure that each reference is complete with citation year, volume, and page/locator numbers; and that correct and congruent publication years are used in the text and in the reference list (e.g. currently discrepancy for Chhatre et al 24 vs 25).

- As we are switching from a free-text author contribution statement towards a more formal statement based on Contributor Role Taxonomy (CRediT) terms, please remove the present Author Contribution section and instead specify each author's contribution(s) directly in the Author Information page of our submission system during upload of the final manuscript. See <https://casrai.org/credit/> for more information.

- Please adjust the format of the Data Availability section along the lines stipulated in our Guide to Authors: <https://www.embopress.org/page/journal/14602075/authorguide#dataavailability> - including direct URLs to the respective repositories. Suggested wording: "The [structural coordinates | microarray | mass spectrometry] data from this publication have been deposited to the [name of the database] database [URL] and assigned the identifier [accession | permalink | hashtag]." Also, please note that only newly derived/deposited datasets should be listed here. For previously reported/deposited datasets, please mention them were appropriate (e.g. in the Methods section), and consider including formal Data Citations (explained at: <https://www.embopress.org/page/journal/14602075/authorguide#referencesformat>)

- Please rename the associated movie file into Expanded View movies (in-text callouts: "Movie EV1/2"). Their legends should be moved out of the main text into individual text files, each of which should be combined with the respective movie file into a separate ZIP file and uploaded as such.

- During routine pre-acceptance checks, our data editors noted that the measure of center for the error bars needs to be defined in the legend of figure 2E.

- Finally, please provide suggestions for a short 'blurb' text prefacing and summing up the conceptual aspect of the study in two sentences (max. 250 characters), followed by 3-5 one-sentence 'bullet points' with brief factual statements of key results of the paper; they will form the basis of an editor-written 'Synopsis' accompanying the online version of the article. As "visual title" for the synopsis section of your paper, I would probably simply use the schematics from Figure 6, which should nicely fit in terms

of aspect ratio and resolution.

I am returning the manuscript to you for a minor revision for making these modifications, hoping you will be able to swiftly upload the final version, latest by early next week, in order to ensure timely acceptance and publication still within this year.

With kind regards,
Hartmut

9) To facilitate reproducibility and cross-laboratory adoption of methodologies, please structure the Materials & Methods section as outlined in our guide to authors, including a completed Reagents and Tools Table that can be downloaded from our author guidelines as well (<https://www.embopress.org/page/journal/14602075/authorguide#structuredmethods>).

10) Digital image enhancement is acceptable practice, as long as it accurately represents the original data and conforms to community standards. If a figure has been subjected to significant electronic manipulation, this must be clearly noted in the figure legend and/or the 'Materials and Methods' section. The editors reserve the right to request original versions of figures and the original images that were used to assemble the figure. Finally, we generally encourage uploading of numerical as well as gel/blot image source data; for details see: embopress.org/page/journal/14602075/authorguide#sourcedata

In the interest of ensuring the conceptual advance provided by the work, we recommend submitting a revision within 3 months (4th Feb 2026). Please discuss the revision progress ahead of this time with the editor if you require more time to complete the revisions. Use the link below to submit your revision:

Link Not Available

Referee #2:

The authors revised the manuscript to make their experimental and computational results fitted to the conclusions. With this revision, the logic of the paper is more consistent than the previous version. Regarding a major point from this and another reviewers regarding the role of tyrosination for dynein-2 to target the A-tubule, the authors mentioned that the VashL mutant, which has high degrees of tyrosination, prevents dynein-2 binding, while they did not show the data due to time limitation for revision. Since this data seems clear evidence of tyrosination-based dynein-2 binding regulation, this reviewer would strongly recommend the author to conduct the experiment, or any other experiment Reviewer #1 suggests. It would be great if the editor considers prolongation of the deadline to allow the authors to conduct an experiment to prove the role of tyrosination. Other points are somehow addressed. While quantities of tyrosinated tubulins in the A- and B-tubules respectively were not given, this reviewer agrees that it is a very challenging demand.

Referee #3:

The authors have substantially revised their manuscript in response to the initial round of reviews, and the work is now significantly strengthened. I am pleased to recommend it for publication. The study addresses the important question of how the IFT motor dynein-2 is sorted to its correct track, the A-tubule of the ciliary doublet. The combination of cryo-ET, biochemical data, and MD simulations now presents a cohesive and mechanistically insightful model where this preference is tuned by the tyrosination state of α -tubulin, representing a notable advance for the field.

My primary concern in the initial review was the lack of direct, mechanistic evidence for how tyrosination mediates binding. The authors have addressed this concern effectively. While they reasonably explain the technical limitations preventing high-resolution 3D reconstructions from their cryo-ET data, they have provided compelling new analyses that offer the required insight. The new contact map analysis from their MD simulations is a particularly strong addition, providing a concrete, residue-level model of the binding interface. This, combined with the new quantitative mass spectrometry data and the statistical validation of dynein-2's binding preference, provides the necessary rigor to support their claims.

While the authors have not performed the definitive experiment to directly prove the role of tyrosination, they have successfully compensated for this by substantially strengthening their manuscript in multiple other areas. The collective weight of the new experimental and computational evidence now provides a compelling case for their model. The paper is much improved, and I have no further concerns.

Point-by-point response for editorial issues

Thank you very much for the helps from the editorial office. We have fixed the files as below and highlighted in cyan to distinguish from the places fixed according to the reviewers' comments (highlighted in yellow).

- Please upload all main Figures and all Expanded View figures as image rather than PPTX files, with sufficient resolution/quality for production.

We have uploaded the tiff file version of Figure 4 instead of PPTX file.

- Please adjust the order of the manuscript sections, and also make sure to use the correct section headers:

Title page with complete author information, Abstract, Keywords, Introduction, Results, Discussion, Methods, Data Availability, Acknowledgements, Disclosure and Competing Interests Statement, References, Main Figure Legends, Tables, Expanded Figure Legends. Please remove the duplicated "competing interest" statement from the title page.

We have fixed the order of the manuscript sections, and deleted the duplicated competing interest statement from the title page. Fixed points are highlighted in cyan.

- On the abstract page of the manuscript, please include 4-5 general keyword terms to enhance searchability.

We included keywords related to our work in the abstract page (Lines 34-35).

- Please carefully go through the reference list and make sure that each reference is complete with citation year, volume, and page/locator numbers; and that correct and congruent publication years are used in the text and in the reference list (e.g. currently discrepancy for Chhatre et al 24 vs 25).

We have double checked all the references and fixed the citations with mistakes (Lines 289, 296, and 733, and Reference section) .

- As we are switching from a free-text author contribution statement towards a more formal statement based on Contributor Role Taxonomy (CRediT) terms, please remove the present Author Contribution section and instead specify each author's contribution(s) directly in the Author Information page of our submission system during upload of the final manuscript. See <https://casrai.org/credit/> for more information.

We have removed the Author Contribution section from the main text.

- Please adjust the format of the Data Availability section along the lines stipulated in our [Guide](https://www.embopress.org/page/journal/14602075/authorguide#dataavailability) to [Authors: https://www.embopress.org/page/journal/14602075/authorguide#dataavailability](https://www.embopress.org/page/journal/14602075/authorguide#dataavailability) - including direct URLs to the respective repositories. Suggested wording: "The [structural coordinates | microarray | mass spectrometry] data from this publication have been deposited to the [name of the database] database [URL] and assigned the identifier [accession | permalink | hashtag]."
Also, please note that only newly derived/deposited datasets should be listed here. For previously reported/deposited datasets, please mention them were appropriate (e.g. in the Methods section), and consider including formal Data Citations (explained at: <https://www.embopress.org/page/journal/14602075/authorguide#referencesformat>)

We have modified the Data availability section accordingly (Lines 668-670).

- Please rename the associated movie file into Expanded View movies (in-text callouts: "Movie EV1/2"). Their legends should be moved out of the main text into individual text files, each of which should be combined with the respective movie file into a separate ZIP file and uploaded as such.

We have fixed the in-text callouts (Lines 160 and 911), and we have removed the legends from main text, and uploaded the ZIP files containing legends as text files and corresponding movie files.

- During routine pre-acceptance checks, our data editors noted that the measure of center for the error bars needs to be defined in the legend of figure 2E.

We apologize for the lack of the information. We have added the proper sentences (Lines 886-887).

- Finally, please provide suggestions for a short 'blurb' text prefacing and summing up the conceptual aspect of the study in two sentences (max. 250 characters), followed by 3-5 one-sentence 'bullet points' with brief factual statements of key results of the paper; they will form the basis of an editor-written 'Synopsis' accompanying the online version of the article. As "visual title" for the synopsis section of your paper, I would probably simply used the schematics from Figure 6, which should nicely fit in terms of aspect ratio and resolution.

We have uploaded a text for Synopsis.

We have prepared the image explaining the narrative of the paper and have uploaded it. Please check if it meets the requirements for the visual title.

Apart from the editorial comments, this minor point was fixed.

-We have fixed the Fig. 3E since we have noticed one of the insets was missing the scale bar.

-Higher resolution version file of Fig. 5 has been uploaded.

-The label of Fig. EV4 was missing in the figure file, so we have added.

-Some of the callouts of figures were missing a period after "Fig", so we made them consistently written as "Fig.".

- All PDB accession numbers were standardized to uppercase.

We appreciate the editorial staffs again for the assistance.

Dr. Muneyoshi Ichikawa
Fudan University
Department of Biochemistry and Biophysics
2005 hao, Songhu-lu, Yangpu-qu
Shanghai, Shanghai 200438
China

10th Nov 2025

Re: EMBOJ-2025-120686R1

Cryo-ET and MD simulations reveal that dynein-2 is tuned for binding to the A-tubule of the ciliary doublet

Dear Muneyoshi,

Thank you for submitting your final revised manuscript for our consideration. I am pleased to inform you that we have now accepted it for publication in The EMBO Journal.

With kind regards,

Hartmut

Please note that it is The EMBO Journal policy for the transcript of the editorial process (containing referee reports and your response letters) to be published as an online supplement to each paper. If you should prefer removal of any referee-only figures included in the point-by-point response(s), e.g. because they may still be used for future publication or because they have been reproduced from published work by others, please do let us know immediately via response email.

More information is available here: https://www.embopress.org/transparent-process#Review_Process